

# A graph-theoretic approach to chaos and complexity in quantum systems

Maxwell West[1], Neil Dowling[2], Angus Southwell[3], Martin Sevior[1],
Muhammad Usman[1,3,4], Kavan Modi[3] and Thomas Quella[5]

**1** School of Physics, The University of Melbourne, Parkville, VIC 3010, Australia
**2** Institut für Theoretische Physik, Universität zu Köln,
Zülpicher Strasse 77, 50937 Köln, Germany
**3** School of Physics & Astronomy, Monash University, Clayton, VIC 3800, Australia
**4** Data61, CSIRO, Clayton, 3168, VIC, Australia
**5** School of Mathematics and Statistics, The University of Melbourne,
Parkville, VIC 3010, Australian

⋆ westm2@student.unimelb.edu.au , † ndowling@uni-koeln.de ,
‡ thomas.quella@unimelb.edu.au

## Abstract

There has recently been considerable interest in studying quantum systems via *dynamical Lie algebras* (DLAs) – Lie algebras generated by the terms which appear in the Hamiltonian of the system. However, there are some important properties that are revealed only at a finer level of granularity than the DLA. In this work we explore, via the *commutator graph*, average notions of scrambling, chaos and complexity over ensembles of systems with DLAs that possess a basis consisting of Pauli strings. Unlike DLAs, commutator graphs are sensitive to short-time dynamics, and therefore constitute a finer probe to various characteristics of the corresponding ensemble. We link graph-theoretic properties of the commutator graph to the out-of-time-order correlator (OTOC), the frame potential, the frustration graph of the Hamiltonian of the system, and the Krylov complexity of operators evolving under the dynamics. For example, we reduce the calculation of average OTOCs to a counting problem on the graph; separately, we connect the Krylov complexity of an operator to the module structure of the adjoint action of the DLA on the space of operators in which it resides, and prove that its average over the ensemble is lower bounded by the average shortest path length between the initial operator and the other operators in the commutator graph.

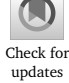
# 1  Introduction

The dynamical Lie algebra (DLA) has recently risen to prominence as an extremely useful characteristic of parametrized quantum systems [1–12], finding applications for example in quantum control [1, 2], simulation [3] and quantum machine learning [4–8]. Given an ensemble of dynamics implementing unitaries that collectively form a Lie group $G$, the DLA is the Lie algebra $\mathfrak{g}$ associated to $G$ via the famous Lie group / algebra correspondence which identifies $\mathfrak{g}$ as the tangent space of $G$ at the identity element. As is prototypically the case in applications of Lie theory, various questions concerning the ensemble $G$ can be translated to questions formulated in terms of $\mathfrak{g}$, where the analysis is facilitated by the availability of the powerful tools of linear algebra. Despite the widespread usefulness of the DLA, however, ensembles of quantum systems exhibit properties of interest that it fails to capture – i.e., and as we shall explore, there are non-trivially distinct dynamics whose DLAs are isomorphic when thought of as abstract Lie algebras, or even *equal* as concrete Lie subalgebras of $\mathfrak{gl}(d)$. Naturally, this encourages the exploration of objects that capture a finer structure than the DLA alone. To that end, in this paper we work with a related object, the *commutator graph*, which represents the adjoint action of a set $\mathcal{G}$ of generators $\{H_\ell\}_\ell$ of a Pauli DLA (a DLA possessing a basis consisting of Pauli strings [6, 11]) on the space of operators on the Hilbert space of the system (see Fig. 1). Importantly, the commutator graph depends explicitly on the choice of generating set $\mathcal{G}$, not merely on the full Lie algebra $\mathfrak{g} = \langle i\mathcal{G}\rangle_{\text{Lie}}$. As we shall discuss, the generating set encodes the short-time dynamics of the ensemble, whereas the full DLA captures *only* the long-time dynamics. While understanding of the short-time behaviour may be extrapolated to long-times, the converse is not necessarily true. Indeed, the DLA is insensitive to ensembles whose average behavior differs only on short timescales.

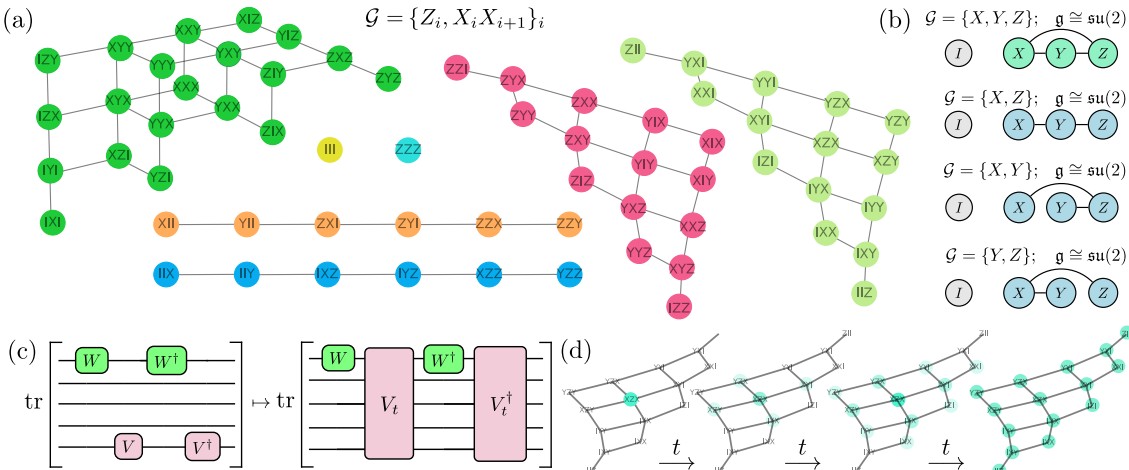

Figure 1: (a) The *commutator graph* of a family of generators $\mathcal{G} = \mathrm{span}_{\mathbb{R}}\{H_\ell\}_\ell$ (with each $H_\ell$ a Pauli string) possesses a node for each $n$-qubit Pauli string, with edges connecting vertices $p$ and $q$ that are linked by the adjoint action of a generator $H_\ell$ of the dynamics, i.e. $[H_\ell, p] \propto q$. The graph breaks up into connected components corresponding to representations of the adjoint action of the *dynamical Lie algebra* (DLA) $\mathfrak{g} = \langle \mathcal{G} \rangle_{\mathrm{Lie}}$ on the space of linear operators $\mathcal{L} = \mathrm{End}\,\mathcal{H}$ on the Hilbert space $\mathcal{H}$ of the system. (b) Importantly, the DLA does *not* fix the graph structure, which depends explicitly on $\mathcal{G}$. While different sets of generators can produce (under the Lie closure) identical DLAs, they do not lead to identical commutator graphs; indeed the graphs are not even necessarily isomorphic. All that the DLA does specify is the set of connected components. (c) The out-of-time-order correlator (OTOC) $F(W, V_t) = d^{-1}\mathrm{tr}[W V_t W^\dagger V_t^\dagger]$ gives a common diagnostic of scrambling in quantum systems: the exponential decay of the OTOC between two initially commuting operators. Properties of the commutator graph allow us to evaluate the average OTOC between Pauli strings for a wide class of dynamics. (d) An operator which is initially a Pauli string will spread across the connected component of the graph to which it belongs via Heisenberg time evolution. The connected components therefore contain the *Krylov spaces* of their constituent nodes. At long times, the details of the internal edge structure of a connected component on average "wash out", and all of the remaining information is captured at the level of the DLA.

In this work, we leverage the properties of the commutator graph for the study of many-body systems, proving a sequence of results relating its graph-theoretic characteristics to information scrambling [13–17], Krylov complexity [18–20], classical simulability [3], and the frustration graph of the Hamiltonian underlying the dynamics [21]. At a high level, our results mainly concern average properties of operators Heisenberg-evolving under a unitary sampled from a given class of dynamics. In particular, we investigate their tendency to scramble, as probed by the out-of-time-order correlator (OTOC) [13–15]. Operator scrambling is a well-investigated phenomenon in many-body physics, with the OTOC often touted as a quantum analogue of the Lyapunov exponent [16], and its exponential decay being a necessary condition for quantum chaos [17]. More recently, scrambling has been shown to influence the performance [22–24] and robustness [25] of quantum machine learning models. By connecting the scrambling properties of classes of Hamiltonians to characteristics of the commutator graph – which we can analyse from the perspective of representation theory – we are able to derive a sequence of broadly applicable results. To give a few examples of the flavour of these

results, we find (e.g.) that the (second-order) *frame potential* of an ensemble [26] can be expressed as the product of the number of components of its commutator graph and the number of isolated vertices it contains; in another result, we find that the average OTOC between two Pauli strings can be related to the fraction of strings in the connected component of one with which the other fails to commute.

On the complexity side we introduce a new measure of operator complexity that we dub *graph complexity*. The graph complexity of a Heisenberg-evolved Pauli string $p_t = e^{iHt} p e^{-iHt}$ measures the weighted distance (as measured by the shortest path between nodes of the commutator graph) between the Pauli strings that appear in the expansion of $p_t$ and $p$ itself. We prove that graph complexity belongs to the class of *q-complexities* introduced in Ref. [18], and therefore lower bounds the *Krylov complexity* of $p_t$. As we shall see, graph complexity is an example of a quantity that is not constant between isomorphic DLAs (but rather explicitly depends on the chosen generating sets), and is therefore captured only at a level of detail at least as fine-grained as that of the commutator graph.

We begin by reviewing some necessary background knowledge in Section 2, covering both the construction of the DLA and the commutator graph, as well as some concepts from many-body physics which will be employed, before covering our analytical results in Section 3. An example-based exploration of various dynamics is given in Section 4, before we conclude in Section 5.

## 2 Preliminaries

### 2.1 The commutator graph

Throughout this work we will be interested in studying ensembles of dynamics generated by Hamiltonians of the form

$$H = \sum_{p \in \mathcal{G}} c_p p \,, \tag{1}$$

where $\mathcal{G}$ is a fixed, family-defining set of Pauli strings. One can associate to such a family a *dynamical Lie algebra*

$$\mathfrak{g} = \langle i\mathcal{G} \rangle_{\text{Lie}} \,, \tag{2}$$

the smallest real Lie algebra containing $i\mathcal{G}$ [1,4]. Algorithmically, this is given by the real span of the repeated nested commutators of ($i$ times) the elements of $\mathcal{G}$,

$$\mathfrak{g} = \text{span}_{\mathbb{R}} \big[ i\mathcal{G} \cup [i\mathcal{G}, i\mathcal{G}] \cup [i\mathcal{G}, [i\mathcal{G}, i\mathcal{G}]] \cup \dots \big] \,. \tag{3}$$

For physically relevant Hamiltonians the Pauli strings in the set of generators $\mathcal{G}$ are usually local, while the iterated commutators will generally produce highly non-local terms. Alternatively, from a quantum computing point of view, given a quantum circuit

$$U(\vartheta) = \prod_{l=1}^{L} \prod_{p \in \mathcal{G}}^{K} e^{-i\vartheta_{lp} p} \,, \tag{4}$$

consisting of repeated layers of unitaries generated by $\mathcal{G}$ and implementing via Trotterisation the dynamics associated with the Hamiltonian (1) (by making the choice $\vartheta_{lp} = c_p t/L$), $\mathfrak{g}$ captures the expressibility of the circuit in the sense that every unitary of the form of Eq. (4) is generated by an element of the DLA, i.e. $\forall \vartheta \; \exists X \in \mathfrak{g}$ such that $U(\vartheta) = \exp(X)$ (which can be seen, for example, by the Baker-Campbell-Hausdorff formula). Such unitaries lie in a Lie group corresponding to $\mathfrak{g}$ (which throughout this work we shall denote by $G$). In this way we

Table 1: **Dynamical Lie Algebras and Commutator Graphs.** Here we collect a representative sample of dynamical Lie algebras, and record some key characteristics of their corresponding commutator graphs. The DLAs corresponding to 1D translationally invariant 2-local spin systems (and indeed spin systems on undirected graphs [27]), upon which we primarily focus, have been completely classified [9], with dimensions scaling either linearly, quadratically or exponentially with the system size $n$. The product #(single element components)#(components) gives the $k = 2$ frame potential, $F_G^{(2)}$ [26, 28]. $F_G^{(2)}$ attains its minimal value of 2 exactly if sampling Haar randomly from $G$ produces a 2-design; higher values of $F_G^{(2)}$ reflect increasingly significant deviations from being a 2-design. *In many cases, the number of graph components coincides with the number of module components, defined as the number of irreducible modules appearing in the decomposition of the adjoint representation with respect to the DLA. Where this is not the case, the number of module components is written in brackets. The mismatch arises when the irreducible modules do not all admit a basis of Pauli strings.

| Model | Generators | Lie Algebra | Dimension | Graph (Module) Components* | Isolated Vertices | Component Dimensions |
|---|---|---|---|---|---|---|
| Matchgate | $Z, XX$ | $\mathfrak{so}(2n)$ | $n(2n-1)$ | $2n+1$ $(2n+2)$ | 2 | $\binom{2n}{\kappa}$, $\kappa = 0, 1, 2, \ldots, 2n$ |
| Universal | $X, Y, ZZ$ | $\mathfrak{su}(2^n)$ | $4^n - 1$ | 2 | 1 | $1, 4^n - 1$ |
| XY model+$B_X$ | $XX, YY, X$ | $\mathfrak{su}(2^{n-1})^{\oplus 2}$ | $2^{2n-1} - 2$ | 6 (8) | 2 | $1, 1, 4^{n-1}-1, 4^{n-1}-1, 4^{n-1}, 4^{n-1}$ |
| Ising model+$\vec{B}$ | $XX, X, Y, Z$ | $\mathfrak{su}(2^{n-1})^{\oplus 2} \oplus \mathfrak{u}(1)$ | $2^{2n-1} - 1$ | 6 (8) | 2 | $1, 1, 4^{n-1}-1, 4^{n-1}-1, 4^{n-1}, 4^{n-1}$ |
| Orthogonal | $XY, YX, YZ, ZY$ | $\mathfrak{so}(2^n)$ | $2^{n-1}(2^n-1)$ | 3 | 1 | $1, 2^{n-1}(2^n-1), 2^{n-1}(2^n+1)-1$ |
| Symplectic | $X_1, Z_1 Z_2,$ $Y_i, X_i Y_{i+1}, Y_i X_{i+1}$ | $\mathfrak{sp}(2^{n-1})$ | $2^{n-1}(2^n+1)$ | 3 | 1 | $1, 2^{n-1}(2^n+1), 2^{n-1}(2^n-1)-1$ |

associate a Lie algebra $\mathfrak{g}$ and a Lie group $G$ to families of Hamiltonians, the members of which share the same set of generators (or interaction terms) but differ in their relative coefficients.

In this work we will exclusively consider *Pauli string DLAs* [6, 11], namely DLAs that have a generating set consisting entirely of Pauli strings. This is not too terrible a restriction, as in practice the Hamiltonians and quantum circuits one works with are often of this form, and provide a wide class of interesting examples. More generally, one could choose a different basis $\{O_i\}_{i=1}^{\dim \mathcal{L}}$ for the operator space $\mathcal{L} = \text{End}(\mathcal{H})$, and focus on DLAs that possess as a basis a subset of the basis of $\mathcal{L}$. Subject to the condition that the elements of the basis are closed (up to scalar multiplication) under commutation, i.e. $\forall\, 1 \le i, j \le \dim \mathcal{L}\ \exists k : [O_i, O_j] \propto O_k$, one can give a definition of the commutator graph analogous to the one we will present in the Pauli case, upon which our focus is informed both by its physical relevance and its advantage of comprising (operator-entanglement-free) product operators with algebraically tractable commutation relations.

The commutator graph was introduced in Ref. [6], where it was used to characterise the trainability of QML models constructed from a free-fermionic system. In that application, it facilitated the study of operators which are not themselves within the DLA, which had been a restriction of several preceding works [4, 5]. The commutator graph of an $n$-qubit system consists of $4^n$ vertices, with one corresponding to each Pauli string. Two vertices, corresponding to Pauli strings $p$ and $q$, are connected by an edge if $\exists X \in \mathcal{G}$ such that $[p, X] \propto q$. We note that the commutator of two Pauli strings is always either zero or ($\pm 2i$ times) another Pauli string, and that, separately, $[p, X] \propto q \implies [q, X] \propto p$; the commutator graph is therefore

*undirected*. Importantly, isomorphic DLAs need *not* induce isomorphic graphs; for example consider $\mathfrak{g}_1 = \mathrm{span}_{\mathbb{R}}\{iI\}$ and $\mathfrak{g}_2 = \mathrm{span}_{\mathbb{R}}\{iX\}$ which are isomorphic as (abelian) Lie algebras, but will clearly induce non-isomorphic commutator graphs. Moreover, having fixed a choice of computational basis, even DLAs that are *equal* as explicit Lie algebras spanned by a common set of Pauli strings can lead to different commutator graphs, as the construction explicitly depends on the choice of generators (see Fig. 1(b)). From a representation-theoretic point of view, the first of these points may be explained by the well-known fact that there can exist multiple non-isomorphic embeddings of a Lie algebra into a larger Lie algebra. They may for instance differ in their "embedding index" (see [29] and references therein). For example, consider the case $\mathfrak{su}(2) \subset \mathfrak{su}(3)$. One can explicitly embed $\mathfrak{su}(2)$ into $\mathfrak{su}(3)$ by placing the matrices of $\mathfrak{su}(2)$ into the "upper $2 \times 2$ block" of $\mathfrak{su}(3)$; in this case the defining representation of $\mathfrak{su}(3)$ on $\mathbb{C}^3$ decomposes into the defining (spin-$\frac{1}{2}$) representation of $\mathfrak{su}(2)$ plus a singlet (corresponding to the trivial action on the last component of $\mathbb{C}^3$). Alternatively, one can use the equivalence $\mathfrak{su}(2) \cong \mathfrak{so}(3)$ for the embedding; in this case, the defining representation of $\mathfrak{su}(3)$ decomposes into the spin-1 representation of $\mathfrak{su}(2)$ (which corresponds to the defining representation of $\mathfrak{so}(3)$).

In general, the commutator graph breaks up into connected components corresponding to modules of the adjoint action of $G$ (see Fig. 1(a)), with an initial Pauli string $p$ being mapped under the dynamics to $p_t = e^{iHt} p e^{-iHt}$, a linear combination of Pauli strings in the same connected component as $p$ (see Fig. 1(d)). This follows from the Baker-Campbell-Hausdorff formula and $G = \exp(\mathfrak{g})$; $p_t$ is obtained from $p$ by taking a linear combination of repeated nested commutators of elements of the DLA, leading to precisely the Pauli strings contained within the same component of the graph as $p$. The Pauli strings spanning $\mathfrak{g}$ itself will appear as a single component if $\mathfrak{g}$ is a simple Lie algebra; more generally the various simple summands of $\mathfrak{g}$ will appear as separate components. For example, the (simple) matchgate DLA $\mathfrak{g} \cong \mathfrak{so}(2n)$ appears as a single light green component in Fig. 1(a).

The majority of our results will rely on the evaluation via Weingarten calculus [26] of integrals of the form

$$\mathcal{T}_G^{(2)}(M) = \int_G d\mu_G(U)\, U^{\dagger \otimes 2} M U^{\otimes 2}\,, \tag{5}$$

where $\mu_G$ is the Haar measure on the (compact) Lie group $G$ (possessing the Pauli Lie algebra $\mathfrak{g}$) and $M \in \mathrm{End}(\mathcal{H}^{\otimes 2})$ is an operator on the doubled space $\mathcal{H}^{\otimes 2}$. It is well-known that such integrals correspond to an orthogonal projection of $M$ onto the second-order commutant of $G$, i.e. the space of operators $\{Q : [Q, U^{\otimes 2}] = 0\ \forall U \in G\}$ [26]; obtaining a basis for this space of *quadratic symmetries* is therefore the key step to evaluating Eq. (5).

Before tackling this, let us say a little about the *linear* symmetries of $G$, i.e. the set $\{L : [L, U] = 0\ \forall U \in G\}$. One can think of the linear symmetries as corresponding precisely to the singlets (trivial representations) of $\mathcal{L} = \mathcal{H} \otimes \mathcal{H}^*$; indeed, note that $[L, U] = 0 \iff U^{\dagger} L U = L$. Given a decomposition

$$\mathcal{H} = \bigoplus_{\lambda_0} \bigoplus_{i_0=1}^{m_{\lambda_0}^{\mathcal{H}}} \mathcal{H}_{\lambda_0, i_0}\,, \tag{6}$$

of $\mathcal{H}$ into irreducible $\mathfrak{g}$-modules (with multiplicities counted by the second index), by Schur's lemma such singlets occur (once) for every pairing $(\mathcal{H}_{\lambda_0, i_0}, \mathcal{H}_{\lambda_0, j_0}^*)$ in $\mathcal{L}$ of an irrep of $\mathcal{H}$ with the dual of an irrep isomorphic to it. Explicitly, these symmetries are given by the isomorphisms

$$L_{\lambda_0, i_0, j_0} = \sum_{\alpha_0} |\lambda_0, i_0, \alpha_0\rangle\langle \lambda_0, j_0, \alpha_0|\,, \tag{7}$$

where we have introduced bases $\{|\lambda_0, i_0, \alpha_0\rangle\}_{\alpha_0=1}^{d_{\lambda_0}^{\mathcal{H}}}$ and $\{|\lambda_0, j_0, \alpha_0\rangle\}_{\alpha_0=1}^{d_{\lambda_0}^{\mathcal{H}}}$ of $\mathcal{H}_{\lambda_0,i_0}$ and $\mathcal{H}_{\lambda_0,j_0}$ respectively, whose elements (for any shared value of $\alpha_0$) are mapped to one another under an isomorphism between the two spaces considered as $\mathfrak{g}$-modules.

Returning to the study of Eq. (5), it turns out that, in the case of a Pauli string DLA, if we have a Pauli string basis $\{L_j\}_j$ of the linear symmetries of $G$, the commutator graph allows us to construct a basis of the quadratic symmetries as the elements [6]

$$Q_{j,\kappa} = \sum_{S \in C_\kappa} S \otimes L_j S, \tag{8}$$

where the sum is over Pauli strings $S$ in a given connected component $C_\kappa$ of the graph. A proof of this fact may be found in Ref. [6]. Evaluating integrals of the form of Eq. (5) then reduces to simply computing the projection of $M$ onto the span of the $\{Q_{j,\kappa}\}$, and is a process which we will carry out repeatedly in this work. We note that the symmetries produced by Eq. (7) will *not* necessarily be Pauli strings (but one may be able to find linear combinations of them that are); we will return to this point when we consider the example of matchgate circuit dynamics in Section 4.

The result of Eq. (8) is reminiscent of the more general representation theoretic description of quadratic symmetries; given a decomposition of $\mathcal{L}$ into $\mathfrak{g}$-irreps,

$$\mathcal{L} = \bigoplus_\lambda \bigoplus_{i=1}^{m_\lambda^{\mathcal{L}}} \mathcal{L}_{\lambda,i}, \tag{9}$$

quadratic symmetries correspond to the trivial irreps in $\mathcal{L}^{\otimes 2}$,[1] and – as a result of Schur's lemma – are obtained exactly once for each pair of isomorphic irreps in the decomposition Eq. (9).[2] Given such a pair $(\mathcal{L}_{\lambda,i}, \mathcal{L}_{\lambda,j})$, with bases $\{B_{\lambda,i,\alpha}\}_{\alpha=1}^{d_\lambda^{\mathcal{L}}}$ and $\{B_{\lambda,j,\alpha}\}_{\alpha=1}^{d_\lambda^{\mathcal{L}}}$ respectively, one obtains a quadratic symmetry

$$\widetilde{Q}_{\lambda,i,j} = \sum_{\alpha=1}^{d_\lambda^{\mathcal{L}}} B_{\lambda,i,\alpha} \otimes B_{\lambda,j,\alpha}^\dagger. \tag{10}$$

Despite the clear similarity to Eq. (8), the representation theoretic and commutator graph perspectives can differ, due to the possibility of the connected components of the graph failing to correspond precisely to irreps of the adjoint action of $\mathfrak{g}$, a point to which we will return later.

Finally, we will occasionally work in the "vectorised" formalism [26] wherein one identifies (via the Choi-Jamiołkowski isomorphism) operators with states on a doubled Hilbert space,

$$O \in \text{End}(\mathcal{H}) \iff |O\rangle\!\rangle := (O \otimes \mathbb{1}_{\mathcal{H}})|\Phi^+\rangle \in \mathcal{H}^{\otimes 2}, \tag{11}$$

with $|\Phi^+\rangle$ a maximally entangled state across the two spaces. Similarly, operators on $\text{End}(\mathcal{H})$ can be represented as "super-operators" on $\text{End}(\mathcal{H}^{\otimes 2})$. For example, in the case of a Pauli DLA with a basis of Pauli linear symmetries, from the above discussion we see that the twirl of Eq. (5) (and its natural first-order analogue) are represented as

$$\mathsf{T}_G^{(1)} = \sum_j |L_j\rangle\!\rangle\langle\!\langle L_j|, \tag{12}$$

and

$$\mathsf{T}_G^{(2)} = \sum_{\kappa,j} |Q_{\kappa,j}\rangle\!\rangle\langle\!\langle Q_{\kappa,j}|, \tag{13}$$

or, more generally, the analogous expressions involving Eqs. (7) and (10).

---

[1]Similarly to the linear case, note $[Q, U^{\otimes 2}] = 0 \iff (U^{\otimes 2})^\dagger Q U^{\otimes 2} = Q$.

[2]We note that $\mathcal{L}$ is self-dual as a module of $\mathfrak{g}$ so that pairs of isomorphic irreps are in one-to-one correspondence with pairs of mutually dual irreps that produce singlets in $\mathcal{L}^{\otimes 2}$. This correspondence is the reason for the occurrence of $B^\dagger$ in (10).

## 2.2 Out-of-time-order correlators and the frame potential

The OTOC – which will feature heavily in our results – is a four-point correlator, defined between two unitary operators $W$ and $V_t$ as [13–15]

$$F(W, V_t) = \frac{1}{d} \operatorname{tr}\left[W^\dagger V_t^\dagger W V_t\right], \tag{14}$$

where $d$ is the total Hilbert space dimension. For two initially local operators with disjoint support (which therefore commute) the OTOC is equal to one; the decay of the OTOC as one of the operators is time-evolved, and therefore begins to fail to commute with the other (see Fig. 1(b)), is a well-studied probe of scrambling in quantum systems [13–15]. In fact, exponential OTOC decay is necessary for the chaotic growth of local-operator entanglement [17], considered to be a faithful indicator of non-integrability [30].

An intriguing property of the OTOC is its connection to the *frame potential* $F_{\mathcal{E}}^{(k)}$ of an ensemble $\mathcal{E}$, defined as [26, 28]

$$F_{\mathcal{E}}^{(k)} = \mathbb{E}_{U,V \sim \mathcal{E}}\left[\left|\operatorname{tr}\left(UV^\dagger\right)\right|^{2k}\right]. \tag{15}$$

The frame potential $F_{\mathcal{E}}^{(k)}$ measures the (2-norm)[3] distance between $\mathcal{E}$ and a $k$-design [26], i.e. how close it is to being indistinguishable, up to the $k$th moment, from unitaries sampled via the Haar measure $\mu_{\mathbb{U}(d)}$ on $\mathbb{U}(d)$. In this work we will employ (with $\mathbb{P}$ the set of all Pauli strings) the characterisation [28]

$$d^4 F_{\mathcal{E}}^{(2)} = \sum_{A_1, B_1, A_2, B_2 \in \mathbb{P}} \left|\int_{\mathcal{E}} d\mu_{\mathcal{E}}(U) \operatorname{tr}\left[A_1 U B_1 U^\dagger A_2 U B_2 U^\dagger\right]\right|^2, \tag{16}$$

to exactly evaluate $F_{\mathcal{E}}^{(2)}$ for any ensemble uniformly sampled from a Lie subgroup $G = \exp\{\mathfrak{g}\}$ of $\mathbb{U}(d)$ (where $\mathfrak{g}$ is generated by a set of Pauli strings) as a function of simple properties of the commutator graph (see Prop. 1). More generally, a similar identification exists between $2k$-point OTOCs evaluated at, and averaged over, the Pauli strings, and $k^{\text{th}}$ frame potentials [28], which probe ever finer details of chaos and complexity; in this work we will focus primarily on $k = 2$.

In fact, the frame potential is also intimately connected with the structure of the tensor product representations of the ensemble, as quantified by the following simple observation: given a decomposition of the $k^{\text{th}}$-order tensor product representation $\mathcal{L}^{\otimes k} : U \in \mathcal{U} \mapsto U^{\otimes k}$ of a unitary ensemble $\mathcal{U}$ equipped with an invariant measure into irreducible components,

$$\mathcal{L}^{\otimes k} = \bigoplus_{\lambda^{(k)}} \bigoplus_{i=1}^{m_{\lambda^{(k)}}} \mathcal{L}_{\lambda^{(k)}, i}, \tag{17}$$

the corresponding $k^{\text{th}}$-order frame potential satisfies

$$F_{\mathcal{E}}^{(k)} = \sum_{\lambda^{(k)}} m_{\lambda^{(k)}}^2. \tag{18}$$

This result follows quickly from the connection between the multiplicities of irreps appearing in a given representation and the norm of the character vector induced by the usual inner product on class functions of a group [33–35] and the fact that $\operatorname{tr}\left[U^{\otimes k}\right] = \operatorname{tr}[U]^k$.

---

[3]One can also choose other ways to measure the distance to the Haar moments, such as the diamond norm or relative error distance [31]; they are however not suitable for out-of-time-ordered quantities such as the OTOC (see e.g. Ref. [32]).

## 2.3 Krylov complexity

Several of our results will explore the interplay between the commutator graph and both Krylov complexity and Krylov spaces, the necessary details of which we briefly recap here. For a more detailed exposition one can see e.g. Refs [18–20].

We begin by recalling that the Heisenberg evolution of operators can be succinctly described in terms of the Liouvillian superoperator $\mathsf{L} = [H, -]$ as $O_t = e^{i\mathsf{L}t}O$. In the vectorised picture, the repeated application of the Liouvillian on an initial (vectorised) operator $|O\rangle\!\rangle$ generates a subspace $\{|O\rangle\!\rangle, \mathsf{L}|O\rangle\!\rangle, \mathsf{L}^2|O\rangle\!\rangle, \ldots\}$ called the Krylov space of the operator. A convenient orthonormal (with respect to be inner product $\langle\!\langle O|O'\rangle\!\rangle = (1/d)\,\mathrm{tr}\left[O^\dagger O'\right]$) basis $\{|O_0\rangle\!\rangle, |O_1\rangle\!\rangle, |O_2\rangle\!\rangle, \ldots\}$ can be defined on this space. We begin by letting $|O_0\rangle\!\rangle = |O\rangle\!\rangle$, $|O_1\rangle\!\rangle = b_1^{-1}\mathsf{L}|O\rangle\!\rangle$, with $b_1 = \sqrt{\langle\!\langle \mathsf{L}O|\mathsf{L}O\rangle\!\rangle}$ a normalisation constant (note $\langle\!\langle O_0|O_1\rangle\!\rangle = \mathrm{tr}\left[O[H,O]\right] = 0$). The rest of the construction proceeds recursively; one defines

$$|A_n\rangle\!\rangle = \mathsf{L}|O_{n-1}\rangle\!\rangle - b_{n-1}|O_{n-2}\rangle\!\rangle\,, \tag{19}$$

and $|O_n\rangle\!\rangle = b_n^{-1}|A_n\rangle\!\rangle$, with $b_n = \sqrt{\langle\!\langle A_n|A_n\rangle\!\rangle}$. Eventually there will come an $n$ with $\langle\!\langle A_n|A_n\rangle\!\rangle = 0$, at which point the process (the *Lanczos algorithm*) terminates. Intuitively, starting from a "simple" (e.g. local) operator $O$, basis elements $O_i$ with higher values of $i$ represents increasingly complex operators having been scrambled by the dynamics. Expanding

$$|O_t\rangle\!\rangle = \sum_n i^n \varphi_n(t)|O_n\rangle\!\rangle\,, \tag{20}$$

in this basis (with the factors of $i$ conventionally inserted to render the $\varphi_n$ real [19]), Krylov complexity is defined as the weighted average

$$\mathsf{K}(O_t) = \sum_n n|\varphi_n(t)|^2\,. \tag{21}$$

Eq. (21) can be interpreted physically as the expectation value of the position of $|O_t\rangle\!\rangle$ on a one-dimensional chain consisting of the Krylov basis elements. We will later return to this interpretation, contrasting it with the spread of $|O_t\rangle\!\rangle$ over the commutator graph. From its construction we see that if $O$ is a Pauli string then its Krylov space is a subspace of the space spanned by the nodes of its connected component in the commutator graph. More generally, an initial operator which is a linear combination of Pauli strings will spread across all of the components on which it initially has support.

Krylov complexity has been shown [18] to upper bound all *q-complexities*, a large class of complexity measures. Following Ref. [18] we define a *q-complexity* of an operator $O_t = e^{i\mathsf{L}t}O$ to be the expectation value $\mathsf{Q}(O_t) = \langle\!\langle O_t|\mathcal{Q}|O_t\rangle\!\rangle$ where $\mathcal{Q}$ is a superoperator satisfying

1. $\mathcal{Q}$ is positive semidefinite, with a spectral decomposition

$$\mathcal{Q} = \sum_a q_a |q_a\rangle\!\rangle\langle\!\langle q_a|\,, \tag{22}$$

   where $q_a \geq 0 \ \forall a$.

2. There exists $M > 0$ such that

$$\langle\!\langle q_a|\mathsf{L}|q_b\rangle\!\rangle = 0\,, \quad \text{if} \quad |q_a - q_b| > M\,, \tag{23}$$

$$\langle\!\langle q_a|O\rangle\!\rangle = 0\,, \quad \text{if} \quad |q_a| > M\,. \tag{24}$$

It is proven in Ref. [18] that for any *q*-complexity one has the bound

$$\mathsf{Q}(O_t) \leq 2M\mathsf{K}(O_t)\,. \tag{25}$$

We will later introduce a new *q-complexity*, which we term *graph complexity*, and use it to turn properties of the commutator graph into lower bounds on the Krylov complexity of the corresponding dynamics.

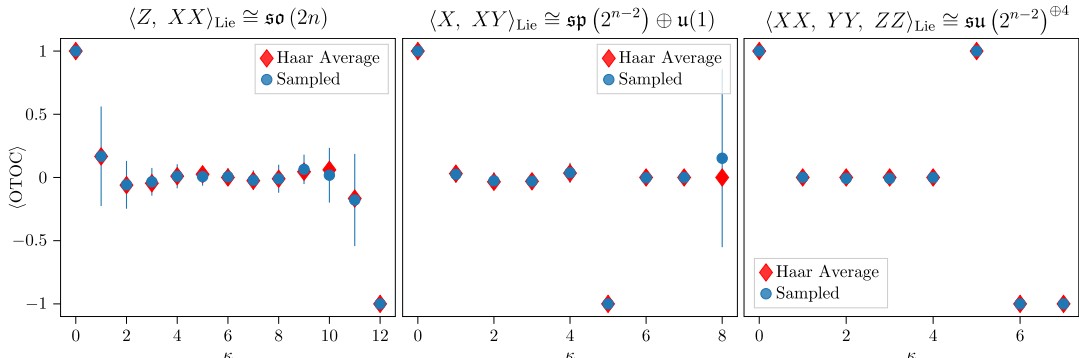

Figure 2: The OTOC between $e^{i\sum_j c_j H_j}(ZXYZIY)e^{-i\sum_j c_j H_j}$ and a representative Pauli string from each connected component of the commutator graph (indexed by $\kappa$, ordered arbitrarily), for 100 (uniformly) random choices of the coefficients $c_j$ appearing in the Hamiltonian. The empirical average value is plotted in blue, with the standard deviation indicated, and compared to the analytical calculation of the Haar average value (in red) given by Prop. 3. Interestingly, we find that this induced distribution well-approximates (at least the second moment of) the uniform distribution, suggesting that our results will with high probability be accurate for Hamiltonians with uniformly randomly sampled coefficients. In small (but non-trivial) components one can find larger variances; for example the "8th" component of the middle dynamics in the (arbitrary) ordering of this figure corresponds to the two-element component consisting of $YXIIII$ and $ZXIIII$.

## 3 Results

We begin by detailing a sequence of results related to (Heisenberg) operator evolution illuminated by the commutator graph. The proofs (which may be found in the appendices) of many of these results ultimately reduce to the fairly mechanical calculation of integrals of the form of Eq. (5) via projection onto the subspace of Eq. (8).

Our first example of an interesting property of an ensemble of unitaries which is determined by graph-theoretic properties of the commutator graph is

**Proposition 1.** *If both the DLA $\mathfrak{g}$ of an ensemble of unitaries $G = \exp \mathfrak{g}$ and its linear symmetries have bases consisting of Pauli strings, then the frame potential $F_G^{(2)}$ can be read off the commutator graph as:*

$$F_G^{(2)} = \#(\text{isolated vertices}) \cdot \#(\text{components}).  \tag{26}$$

In fact, this result follows immediately from the commutator graph based characterisation of the quadratic symmetries combined with the fact that the $k = 2$ frame potential simply counts these symmetries [26] (Eq. (18)). In addition to this, in Appendix A we give an explicit derivation of Eq. (26) via the Weingarten calculus and Eq. (8). This result is consistent with the minimum value of the frame potential being achieved by the ensemble given by Haar randomly sampling from $\mathbb{U}(d)$, in which case the commutator graph splits into $\{\mathbb{1}\} \cup \{\text{non-trivial Paulis}\}$, i.e. two connected components, one of which has a single isolated vertex, and we have $F_{\mathbb{U}(d)}^{(2)} = 2$. In more representation-theoretic language, this reflects the decomposition

$$\mathcal{L} = \text{End}(\mathbb{C}^d) \cong \mathbb{C}^d \otimes (\mathbb{C}^d)^* \cong \mathbb{C} \oplus \text{ad}_{\mathfrak{su}(d)},  \tag{27}$$

of the space of operators on $\mathbb{C}^d$ into a trivial representation $\mathbb{C}$ and the adjoint representation $\text{ad}_{\mathfrak{su}(d)}$ under the action of $\mathfrak{su}(d)$. Here, we interpret $\mathcal{H} = \mathbb{C}^d$ as the fundamental and

$\mathcal{H}^* = (\mathbb{C}^d)^*$ as the anti-fundamental representation of $\mathfrak{su}(d)$.[4] The identity operator will always be in a component by itself as it has a trivial commutator with everything else (corresponding to the representation-theoretic fact that a trivial representation will always arise from the tensor product in Eq. (27) of a representation with its dual), so 2 is clearly the minimum possible value of Eq. (26). More generally, the commutator graph will split into many components, and we will have $F_G^{(2)} \gg 2$. We note that Eq. (27) can also be interpreted in the following way: Under the identification of $\mathrm{End}(\mathbb{C}^d)$ with the (complexification of the) Lie algebra $\mathfrak{u}(d)$, the right hand side corresponds to restricting the adjoint representation of $\mathfrak{u}(d)$ to the action of the Lie subalgebra $\mathfrak{su}(d) \subset \mathfrak{u}(d)$.[5]

Our next result concerns the four-point correlator of Pauli strings $P, Q, R$ and $S$, two of which are Heisenberg-evolved, averaged over the dynamics – this is a generalization of an averaged OTOC, Eq. (14). This expression turns out to be exactly analytically calculable; we find:

**Proposition 2.** *Let $P, Q, R, S$ be Pauli strings belonging respectively to the connected components $C_P, C_Q, C_R, C_S$ of the graph. Then*

$$\mathbb{E}_{U \sim \mu_G} \mathrm{tr}\left[ PUQU^\dagger RUSU^\dagger \right] = d^{-1} \langle\!\langle RP | \mathsf{T}_G^{(1)} | QS \rangle\!\rangle \left( 1 - \frac{2|\{T \in C_Q : \{P, T\} = 0\}|}{|C_Q|} \right). \quad (28)$$

*This expression is zero unless there exists a linear symmetry $L$ such that $RP, QS \propto L$.*

Interestingly, we find a "coherence" effect, where the correlator being non-zero requires that the Pauli strings come from pairs of components that are related by the (multiplicative) action of a linear symmetry. For example, in Fig. 1(a) we find that every component has a "twin" related by the action of $Z^{\otimes n}$ (note that the dark green component is its own twin). As discussed in Appendix A, twin components are isomorphic as graphs, but isomorphic components are not necessarily twins. The coherence effect of Prop. 2 is reminiscent of Theorem 1 of Ref. [6], where the variance of an observable (expressed in terms of its decomposition with respect to a commutator graph) contains a term coming from interactions between two isomorphic modules. That specific example – of "matchgate circuit" dynamics (see Table 1) – is explored in detail in Section 4. In the case where the irreducible $G$-modules can be identified with the connected components of the commutator graph we can interpret the vanishing of the four-point correlator of Prop. 2 when $C_Q \not\cong C_S$ (and, by symmetry, when $C_P \not\cong C_R$) as a consequence of Schur's lemma:

$$\mathbb{E}_{U \sim \mu_G} \mathrm{tr}\left[ PUQU^\dagger RUSU^\dagger \right] = \mathbb{E}_{U \sim \mu_G} \mathrm{tr}\left[ \mathbb{S}(P \otimes R) U^{\otimes 2} (Q \otimes S) U^{\dagger \otimes 2} \right] \quad (29)$$

$$= \mathrm{tr}\left[ \mathbb{S}(P \otimes R) \mathrm{proj}_{(\mathcal{L}^{\otimes 2})^G}(Q \otimes S) \right], \quad (30)$$

as (by Schur's lemma) the trivial irreps that constitute $(\mathcal{L}^{\otimes 2})^G$ come precisely from pairing isomorphic irreps in the decomposition of $\mathcal{L}$. Here, and throughout this work, $\mathbb{S} \in \mathrm{End}\,\mathcal{H}^{\otimes 2}$ denotes the swap operation, i.e. $\mathbb{S}(|\psi\rangle \otimes |\phi\rangle) = |\phi\rangle \otimes |\psi\rangle$.

As a special case of Prop. 2 we can deduce the average OTOC (Eq. (14)) between any two strings, where the Heisenberg-evolution of $V$ is averaged over the dynamics:

**Corollary 3.** *Pick two Pauli strings $V$ and $W$, belonging respectively to the connected components*

---

[4]It is worth noting that the space of operators $\mathcal{L} = \mathcal{H} \otimes \mathcal{H}^*$ is always self-dual as a representation of $\mathfrak{u}(d)$ (or $\mathfrak{su}(d)$). This will then also be true when considered as a representation space with respect to any DLA $\mathfrak{g} \subset \mathfrak{u}(d)$.

[5]A closely related statement is that $\mathfrak{u}(d) = \mathfrak{u}(1) \oplus \mathfrak{su}(d)$, where $\mathfrak{u}(1) = \mathrm{span}_{\mathbb{R}}\{i\mathbb{1}\}$.

*C(V) and C(W) of the graph. Then we have*

$$\underset{U\sim\mu_G}{\mathbb{E}} F(W, U^{\dagger}VU) = 1 - \frac{2|\{T \in C(V) : \{W, T\} = 0\}|}{|C(V)|} \tag{31}$$

$$= 1 - \frac{2|\{T \in C(W) : \{V, T\} = 0\}|}{|C(W)|} . \tag{32}$$

So the average OTOC between two Pauli strings can therefore be calculated as (essentially) just the fraction of nodes in the component of one of them with which the other fails to commute (recalling that Pauli strings either commute or anticommute). We also obtain the following corollary:

**Corollary 4.** *Pick two Pauli strings V and W, belonging respectively to the connected components C(V) and C(W) of the graph. We have:*

$$\frac{|\{T \in C(W) : [V, T] = 0\}|}{|C(W)|} = \frac{|\{T \in C(V) : [W, T] = 0\}|}{|C(V)|} , \tag{33}$$

*and*

$$\frac{|\{T \in C(W) : \{V, T\} = 0\}|}{|C(W)|} = \frac{|\{T \in C(V) : \{W, T\} = 0\}|}{|C(V)|} , \tag{34}$$

*where |S| denotes the number of elements of the set S.*

In the case where the irreps of the adjoint action of $\mathfrak{g}$ on $\mathcal{L}$ possess bases consisting of Pauli strings – so that the connected components of the commutator graph correspond exactly to different irreps – we find that, remarkably, the assignment of Pauli strings to the various irreps is done in such a way as to guarantee that the probability that a given Pauli string $V$ commutes with a randomly chosen Pauli string from the irrep to which $W$ belongs equals the probability that $W$ commutes with a randomly chosen Pauli string from irrep to which $V$ belongs, for all choices of $V$ and $W$.

Corollary 3, which reduces the calculation of average OTOCs to counting the number of Pauli strings from various components of the graph that (anti-)commute with strings from other components allow us to (under certain conditions) connect the average OTOC to the *frustration graph* [21] of the Hamiltonian driving the dynamics. Given a Hamiltonian of the form Eq. (1), the frustration graph possesses a vertex for each Pauli string in the generating set, with an edge connecting a pair of vertices if the corresponding strings anticommute. We emphasise that this is an entirely different graph from the commutator graph that is the main focus of this work. We do have, however:

**Corollary 5.** *If $\mathfrak{g} = \mathrm{span}_{i\mathbb{R}}(\mathcal{G})$ is a simple Lie algebra, then all nodes in the frustration graph of any Hamiltonian of the form Eq. (1) have the same degree.*

We note that a local Hamiltonian will possess $\Theta(n)$ Pauli strings in the decomposition of Eq. (1); their already forming a Lie algebra therefore admittedly limits the applicability of Corollary 5 to a somewhat trivial class of dynamics.

Next we argue that the average OTOC values that we have calculated so far with high probability closely approximate the OTOC that one would obtain for a randomly chosen $U$. Specifically, we have:

**Proposition 6.** *The averages of Proposition 2 and Corollaries 3 and 5 are typical for a classical reductive DLA $\mathfrak{g} = \bigoplus_i \mathfrak{g}_i$ with $\mathfrak{g}_i \in \{\mathfrak{so}(k_i), \mathfrak{su}(k_i), \mathfrak{u}(k_i), \mathfrak{sp}(2k_i)\}$ in the sense that the probability that a given single-shot evaluation with a $U \sim \mu_G$ deviates from the calculated averages is exponentially suppressed,*

$$\mathbb{P}_{U\sim\mu_G}\left(|f(U) - \mathbb{E}[f]| \geq \epsilon\right) \leq 2e^{-(k-2)^2\epsilon^2/384} , \tag{35}$$

*with $k = \min\{k_i\}_i$, and $f(U) = \mathrm{tr}\left[PUQU^{\dagger}RUSU^{\dagger}\right]$, where $P, Q, R$ and $S$ are Pauli strings.*

This result follows from a standard concentration of measure argument, and strengthens the justification of considering only average OTOCs, as we see that for even modest sized systems a given OTOC extremely rarely varies significantly from the average case. Interestingly, we find empirically in Fig. 2 that average OTOCs calculated from the distributions induced by sampling Hamiltonians with uniformly random coefficients also quickly approximate the Haar average values.

For the final result of this section, we return to the dynamics of a single Pauli string. The intuitive picture painted by Fig. 1(d) is that of an operator spreading out to a uniform equilibrium covering the entire component. While this does not necessarily need to be true for a given realisation of the dynamics (i.e. a specific choice of $U \in G$), we do have:

**Proposition 7.** *The average behaviour of an initial Pauli string, when evolved via a random unitary $U \sim \mu_G$, is to spread uniformly over its connected component, i.e. for Paulis $V$ and $W$*

$$\mathop{\mathbb{E}}_{U \sim \mu_G} \left| \left\langle W, U V U^\dagger \right\rangle_{\mathrm{HS}} \right|^2 = \begin{cases} \frac{d^2}{|C(V)|}, & \text{if } C(V) = C(W), \\ 0, & \text{otherwise,} \end{cases} \tag{36}$$

*where $\langle A, B \rangle_{\mathrm{HS}} = \mathrm{tr}\left[ A^\dagger B \right]$ denotes the Hilbert-Schmidt inner product.*

By a similar concentration of measure argument that led to Prop. 6, we have that approximately uniform spreading is what one will typically see.

We now explore the connection between the connected components of the commutator graph and the Krylov space (Sec. 2.3) of an initial Pauli string $p$. Firstly, it is clear from the definition of the graph (not to mention Prop. 7) that the time evolved string $p_t$ will remain within its initial component (say, $C$), and so the dimension of the Krylov space is upper bounded by the number $|C|$ of nodes in the component (as the Pauli strings are orthogonal with respect to the Hilbert-Schmidt inner product used to define the Krylov space). This immediately establishes a link between $|C|$ and the difficulty of classically simulating the evolution of $p_t$; efficient simulations are possible for $|C| = \mathcal{O}(\mathrm{poly}\, n)$, where the evolution is constrained to a tractably small subspace. For example, given a state $|\psi\rangle$ and knowledge of the expectation value $\langle \psi | p | \psi \rangle$ for $q \in C$, the time evolved value $\langle \psi | p_t | \psi \rangle$ can be classically calculated in time polynomial in $|C|$. One can think of this as essentially the "g-sim" algorithm of Ref. [3] (and indeed the earlier works [36,37]), wherein the expectation values of Heisenberg evolving operators within a DLA are seen to be simulable in time $\mathcal{O}(\mathrm{poly}\,(\dim \mathfrak{g}))$; the generalisation to other components of the commutator graph is fairly immediate.

## 3.1 Graph complexity

The commutator graph naturally suggests a new notion of complexity, that quantifies the spread of an initial Pauli string $p$ across its component $C_p$. Specifically, given a $p_t = e^{iHt} p e^{-iHt} = \sum_{q \in C_p} c_q(t) q$, we define its *graph complexity*

$$\mathsf{G}(p_t) = \sum_{q \in C_p} \ell(p, q) |c_q(t)|^2, \tag{37}$$

where we have weighted the expansion coefficient of the Pauli string $q$ by the shortest path length in the graph to the original vertex $p$. Graph complexity is conceptually similar to Krylov complexity (Eq. (21)); whereas Krylov complexity admits an interpretation as the expected position of $p_t$ on the 1-dimensional lattice spanned by the Krylov basis (and therefore the expected distance from the initial state, being localised on the first site) graph complexity is the expected distance between the initial and time-evolved operators as measured by path lengths on the commutator graph. This conceptual similarity can in fact be a numerical equality, as

for example in the case where the initial operator is at the end of a component consisting only of a linear chain (e.g. $ZZY$ in Fig. 1(a)), where the graph complexity and Krylov complexity are identical.

Interestingly, graph complexity fits within the existing landscape of operator complexity measures. Indeed, recalling the definition of $q$-complexity from Sec. 2.3 we arrive at the following result:

**Lemma 8.** *Graph complexity is a q-complexity on the component to which the initial string belongs.*

The proof of this Lemma can be found in Appendix A, where we find that we can take the constant $M$ (that appears in Eqs. (23) and Eqs. (24) of the definition $q$-complexity) to be one. By the general considerations of Ref. [18] this immediately implies that we have $\mathsf{G}(p_t) \leq 2M\mathsf{K}(p_t) = 2\mathsf{K}(p_t)$ (Eq. (25)). In fact, in this instance we can directly prove a tighter (and optimal) bound:

**Lemma 9.** *Krylov complexity constitutes a tight upper bound to the graph complexity.*

As previously mentioned, the bound $\mathsf{G}(p_t) \leq \mathsf{K}(p_t)$ is saturated (for example) by an operator which sits at the end of a linear chain component (see Fig. 1(a)). From Prop. 7 we know that the average behaviour of a Pauli string $p$ undergoing Heisenberg evolution is to spread uniformly across its connected component $C_p$, so that the long time average of the graph complexity is

$$\mathbb{E}_{U \sim \mu_G} \mathsf{G}(U^\dagger p U) = \frac{1}{|C_p|} \sum_{q \in C_p} \ell(p, q). \tag{38}$$

The sum in Eq. (38) is analytically tractable for some graphs; e.g. in the case that $C_p$ forms a linear chain, of which $p$ corresponds to the $j$-th site, a simple calculation gives

$$\mathbb{E}_{U \sim \mu_G} \mathsf{G}(U^\dagger p U) = \frac{j(j+1) + (|C_p| - j)(|C_p| - j + 1)}{2|C_p|}. \tag{39}$$

As a more interesting example, we will later calculate the average graph complexity induced by the "matchgate" dynamics generated by $Z$ and $XX$. We will find that the graph components have sizes $\binom{2n}{\kappa}$ (with $0 \leq \kappa \leq 2n$), but that the graph complexity is upper bounded as $O(n\kappa)$. For $\kappa \sim n$ we have $\binom{2n}{\kappa} \gg n\kappa$; the relative smallness of the graph complexity is a reflection of the (to be seen) existence of short (relative to the size of the component) paths through the component between any pair of vertices.

We can also see that, for short times $t \ll \|H\|_1^{-1}$ (taking $\hbar = 1$ and recalling the Schatten 1-norm $\|H\|_1$ of the Hamiltonian $H$ is the sum of its singular values) graph complexity displays the following universal scaling:

**Lemma 10.** *When $t \ll \|H\|_1^{-1}$ (taking $\hbar = 1$) we have $\mathsf{G}(p_t) = \Theta(t^2)$ (excepting the trivial case of p being a symmetry of the dynamics, in which case $\mathsf{G}(p_t) = 0$ for all t).*

We can also obtain both the constant hidden by the $\Theta$ notation and a simple bound on it; all together we have (seeing the Appendix for the details)

$$\mathsf{G}(p_t) \lesssim t^2 |\mathcal{N}^{(1)}(p)| \|\mathrm{ad}_H(p)\|_1^2 + \mathcal{O}(t^3), \tag{40}$$

where $\mathcal{N}^{(1)}(p)$ denotes we set of nearest neighbours of $p$ in the commutator graph. This universality (up to the dependence on the number of nearest neighbours) is in a sense disappointing; ideally one would hope for different classes of growth rates with time as a function of some interesting property of the dynamics (e.g. integrability). Indeed, this is a key property of Krylov complexity [18].

# 4 Examples

We now apply our results to an illustrative selection of models, tabulated in Table 1. Mostly, we will be interested in determining the commutator graph of a given model and inferring from it what we can of the dynamics; in a few instances we will turn this around, asking what our knowledge of the dynamics can tell us about the commutator graph.

**Universal dynamics.** A simple first example of our framework is provided when the DLA is full rank, i.e. contains all Pauli strings (we may treat DLAs missing only the identity as essentially full rank, lacking only the ability to fix a meaningless global phase). Such a universal DLA may be generated by (for example) $X$, $Y$ and $ZZ$ gates, which yield $\mathfrak{g} \cong \mathfrak{su}(2^n)$ (see Table 1). Now, every simple ideal of $\mathfrak{g}$ that possesses a basis of Pauli strings appears as a connected component of the commutator graph; if $\mathfrak{g}$ is simple and has such a basis (as is and does $\mathfrak{su}(2^n)$) then it appears as a single component. So in this case the commutator graph contains a connected component of size $4^n - 1$; the only remaining Pauli string, the identity, is in a component of its own (as always). From Prop. 1 we obtain that $F^{(2)}_{\mathbb{SU}(d)} = 2$, agreeing (unsurprisingly) with sampling Haar randomly from the full unitary group. We can also calculate the average OTOC between two (non-identity) Pauli strings $V$ and $W$; from Corollary 3 we have

$$\mathop{\mathbb{E}}_{U \sim \mathbb{SU}(d)} F(W, U^\dagger V U) = 1 - \frac{2|\{T \in C(V) : \{W, T\} = 0\}|}{|C(V)|} \tag{41}$$

$$= 1 - \frac{2(2^{2n-1})}{4^n - 1} \tag{42}$$

$$= -\frac{1}{4^n - 1}, \tag{43}$$

using that a (non-identity) Pauli string anticommutes with half of all Pauli strings, all of which live in the large component of the graph. The above integral can of course also be easily evaluated using the usual Weingarten calculus on the unitary group [26], but the counting anticommuting Pauli strings perspective employed here offers some intuition for the perhaps mildly surprising deviance of the answer from zero: the symmetry of (anti-)commuting with exactly half of the Pauli strings is broken (slightly) by the identity residing in a component of its own.

**Free fermions.** A more interesting example is given by the so-called *matchgate circuits*, which correspond (under the Jordan-Wigner transformation) to *fermionic Gaussian unitaries*, i.e. maps between valid representations of *Majorana fermions* [38]. For our purposes such a representation consists of a choice of $2n$ operators $\{c_i\}_{1 \leq i \leq 2n}$ satisfying the canonical anticommutation relations $\{c_i, c_j\} = 2\delta_{i,j}$; for example, one can take

$$\begin{aligned}
c_1 &= XIII \cdots I, & c_2 &= YIII \cdots I, \\
c_3 &= ZXII \cdots I, & c_4 &= ZYII \cdots I, \\
&\vdots & &\vdots \\
c_{2n-1} &= ZZ \cdots ZX, & c_{2n} &= ZZ \cdots ZY.
\end{aligned} \tag{44}$$

The DLA $\mathfrak{g} \cong \mathfrak{so}(2n)$ can be taken to be generated by $Z$ and $XX$; one can show that this is equivalent to taking the linear span of the products of pairwise distinct Majorana modes [6]. In this case the commutator graph decomposes into $2n + 1$ components [6], with dimensions $\binom{2n}{\kappa}$, $\kappa = 0, 1, 2, \ldots, 2n$; the corresponding decomposition $\mathcal{L} = \bigoplus_\kappa \mathcal{L}_\kappa$ can be viewed as the decomposition of $\mathcal{L}$ into spaces corresponding to operators which are a product of $\kappa$ distinct

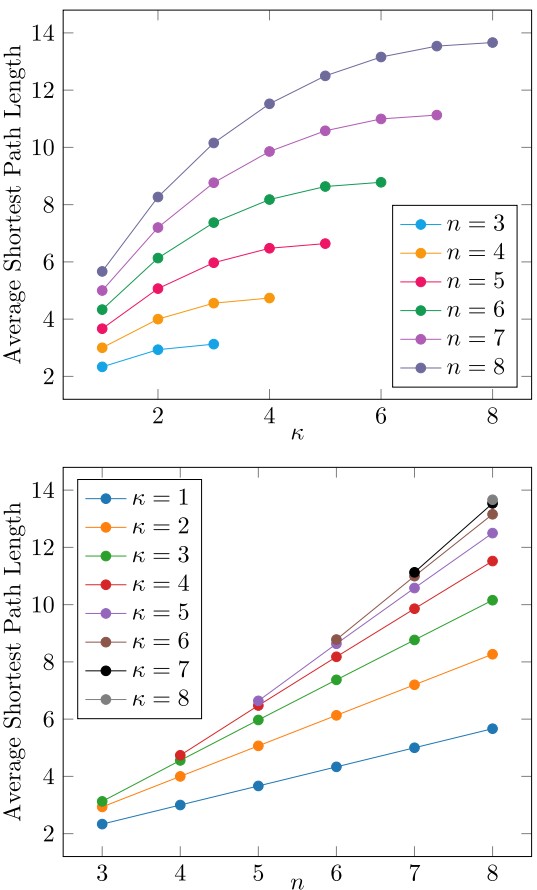

Figure 3: The average path length between vertices in the commutator graph for $Z, XX$ dynamics, as a function of component index $\kappa$ and number of sites $n$. The $\kappa$th component, of dimension $\binom{2n}{\kappa}$, contains the Pauli strings which are a product of $\kappa$ distinct Majorana fermions [6]. For $|\kappa \bmod 2n| \le 2$ the linear growth with the system size $n$ agrees with our analytic calculations; for $|\kappa \bmod 2n| > 2$ the growth also appears to be linear. The isomorphic components with $\kappa > n$ are not shown.

Majoranas [6]. The commutator graph for $n = 3$ is displayed in Fig. 1(a). In particular, note that we have two isolated vertices (i.e. components of dimension one). Representation-theoretically, we can trace this to the fact that the representation of the matchgate circuits on the Hilbert space $\mathcal{H} = \mathbb{C}^d$ with respect to $\mathfrak{so}(2n)$ (where $d = 2^n$) is reducible, and decomposes into two *inequivalent* and mutually dual irreps

$$\mathcal{H} = \mathcal{H}^+ \oplus \mathcal{H}^- \tag{45}$$

(spanned by the computational basis vectors with even and odd Hamming weight respectively). From the discussion of Sec. 2.1 we then expect the space of operators $\mathcal{L} = \mathcal{H} \otimes \mathcal{H}^*$ to contain precisely two trivial irreps,

$$\mathcal{L}^G \cong \left(\mathcal{H}^+ \otimes (\mathcal{H}^+)^*\right)^G \oplus \left(\mathcal{H}^- \otimes (\mathcal{H}^-)^*\right)^G , \tag{46}$$

corresponding to the (linear) symmetries given by

$$\widetilde{L}_1 = \sum_{\substack{x \\ \text{even}}} |x\rangle\langle x| , \qquad \widetilde{L}_2 = \sum_{\substack{x \\ \text{odd}}} |x\rangle\langle x| , \tag{47}$$

where we work in the computational basis and the even (odd) summations are over bitstrings of even (odd) Hamming weight. These symmetries are not Pauli strings, prohibiting their use in of many of our results; happily we can instead use

$$L_1 := \widetilde{L}_1 + \widetilde{L}_2 = I^{\otimes n} \,, \tag{48}$$

and

$$L_2 := \widetilde{L}_1 - \widetilde{L}_2 = Z^{\otimes n} \,. \tag{49}$$

The existence of the linear symmetry $Z^{\otimes n}$ (the "fermionic parity operator" [6]) in this model allows for "coherence" effects between different components, as described in Prop. 2; in particular, they are possible between the isomorphic components corresponding to $\kappa$ and $2n - \kappa$ (for $\kappa \neq n$). This effect was noted and explored in the context of the gradients of variational models defined over matchgate circuits in Ref. [6]. The decomposition of $\mathcal{L}$ into $\mathfrak{g}$-irreps is discussed in Section B. We find the only discrepancy between the commutator graph and representation-theoretic pictures to be that the largest graph component, $\mathcal{L}_n$, decomposes into two inequivalent irreps, corresponding to the $\pm 1$ eigenspaces of $Z^{\otimes n}$. These irreps do not possess Pauli string bases, but are rather spanned by elements of the form $P \pm Z^{\otimes n} P$, and so cannot be distinguished by the commutator graph.

The commutator graph structure can be described very precisely in the Majorana picture. A vertex of the component $C_\kappa$ is described by a choice of $\kappa$ integers $1 \leq i_1 < i_2 < \ldots < i_\kappa \leq 2n$; the corresponding Pauli string is (up to a fourth-root of unity)

$$P = c_{i_1} c_{i_2} \cdots c_{i_\kappa} \,. \tag{50}$$

We show in Appendix C that two such vertices are adjacent in the graph if and only if

$$\sum_{\alpha=1}^{\kappa} |i_\alpha - j_\alpha| = 1 \,. \tag{51}$$

For $\kappa = 1$, then, we obtain a linear chain (the vertices of which correspond to the Majorana modes themselves); for $\kappa = 2$ (the component corresponding to the DLA) the vertices sit at the integer-valued coordinates of an isosceles right triangle of short-side length $2n - 1$ (see Fig. 1(a)). By the above-discussed isomorphism between the components $C_\kappa$ and $C_{2n-\kappa}$, we find analogous behaviour for the components $C_{2n-1}$ and $C_{2n-2}$, respectively. For $\kappa = 1$ the average graph complexity is therefore given by Eq. (39) and is of size $\mathcal{O}(n)$; for $\kappa = 2$ we derive in Appendix C that

$$\begin{aligned}
\mathbb{E}_{U \sim \mu_G} \mathsf{G}(U^\dagger c_{i_1} c_{i_2} U) = \frac{1}{6n(2n-1)} \big( & i_1(i_1 - 1)(6n - i_1 - 1) \\
& + (2n - i_1)(2n - i_1 - 1)(2n - i_1 + 1) \\
& + i_2(i_2 - 1)(i_2 - 2) \\
& + (2n - i_2)(2n - i_2 + 1)(4n + i_2 - 2) \big) \,.
\end{aligned}$$

So again – and despite this component having size $\mathcal{O}(n^2)$ – the asymptotic behaviour is linear in $n$.

More generally, in Appendix C we prove that the average graph complexity of a Pauli string $P$ in the component $C_\kappa$ is upper bounded as

$$\mathbb{E}_{U \sim \mu_G} \mathsf{G}(U^\dagger P U) = O(\kappa n) \,; \tag{52}$$

in Fig. 3 we see numerical evidence of scaling that is indeed linear in $n$.

Finally, by Prop. 1 we have that the frame potential for the ensemble given by uniformly sampling matchgate unitaries is $F^{(2)}_{\text{mgate}} = 2(2n+1) = 4n+2$, growing further from a 2-design for larger systems. From the representation-theoretic point of view we independently calculate $F^{(2)}_{\text{mgate}} = 1^2(2) + 2^2 n = 4n+2$ (Eq. (18)), confirming the result. Unlike in the previous case of universal dynamics, our OTOC results now predict nontrivial differences in behaviour for operators from different connected components. This is demonstrated numerically for $n = 6$ in Fig. 2(a), where an operator is chosen from each component $\mathcal{L}_\kappa$, and its average OTOC with respect to the (pseudorandomly chosen) string $ZXYZIY$ is calculated. In each case we are able to use Corollary 3 to predict the average long-time value, which differ considerably for different choices of the initial string.

**Ising model in an arbitrary field.**    For our next example, we consider the generating set

$$\{X_i\}_{i=1}^n \cup \{X_i X_{i+1}, X_i Y_{i+1}, X_i Z_{i+1}, I_i Y_{i+1}, I_i Z_{i+1}\}_{i=1}^{n-1}\,, \tag{53}$$

corresponding to the Lie algebra

$$\mathfrak{b}_4 \cong \mathfrak{su}(2^{n-1}) \oplus \mathfrak{su}(2^{n-1}) \oplus \mathfrak{u}(1)\,, \tag{54}$$

in the classification of Ref. [9]. By inspection we can identify the $\mathfrak{u}(1)$ summand as corresponding to the span of the central element $X_1$; this is a slightly different situation to the previous example, where we had a linear symmetry $Z^{\otimes n}$ that was not itself in the DLA.

In Appendix B we find a decomposition of $\mathcal{L}$ into $\mathfrak{b}_4$-irreps consisting of subspaces of dimensions $1, 1, 4^{n-1}-1, 4^{n-1}-1, 4^{n-1}$ and $4^{n-1}$. As in the previous example, the commutator graph does not resolve every irrep of $\mathcal{L}$, but rather decomposes into four components, of dimensions $1, 1, 2 \cdot 4^{n-1} - 2$ and $2 \cdot 4^{n-1}$. One can readily see that the two isolated vertices correspond to the linear symmetries $I^{\otimes n}$ and $X_1$, and the two large components respectively to the strings that commute with $X_1$ (less the isolated vertices), and the strings that anticommute with $X_1$. Indeed, consider some Pauli string $P$. It either commutes or anticommutes with $X_1$, a property which it shares with the other vertices in its component,[6] corresponding as they do to strings that are obtained by commuting elements of the DLA (which will commute with the linear symmetry $X_1$) with $P$. Similarly to the component $C_n$ in the matchgate case, we see that the two large components here admit a decomposition into irreps as the $\pm 2i$ eigenspaces of $\text{ad}_{iX_1}$, which are spanned by elements of the form (for Paulis $P$) $P \pm iX_1 P$, whence their invisibility to the commutator graph.

**Orthogonal evolution.**    Next we consider the ensemble of Haar random orthogonal matrices. The applications of random orthogonal matrices to quantum information have been increasingly investigated over the last few years, including for example their uses in randomised benchmarking [34], quantum machine learning [42], classical shadows [43] and as generators of approximate unitary state designs [44]. All of this analysis is facilitated by the fact that, for all $k \geq 1$, an explicit spanning set for the $k^{\text{th}}$ order commutant of the orthogonal group is known [42]; in fact one has that such a set is given by the elements of a certain representation $F_d$ of the *Brauer algebra* $\mathfrak{B}_k(d)$ [39–41], where $d = 2^n$ is again the total dimension of the Hilbert space.[7] One can think of the Brauer algebra as consisting of formal linear combinations of elements of the set of all pairs of a set of $2k$ objects (subsuming therefore the

---

[6]Even without knowing the content of the proceeding sentence, this would just be Corollary 4 with one of the components being of size one.

[7]To avoid possible confusion, we would like to emphasize that the Brauer algebra here is defined as a subalgebra of the space of operators acting on $k$ copies of the *total* Hilbert space on which we have the action of the DLA $\mathfrak{so}(d)$. This is important as the total Hilbert space itself may be viewed as a tensor product, albeit with the DLA not acting on the individual factors.

| | $\sigma$ | $\mathbb{O}$ | $\mathbb{USP}$ |
|---|---|---|---|
| $t=1$ | $\{\{1,2\}\}$ | 1 —— 2 | 1 —— 2 |
| $t=2$ | $\{\{1,3\},\{2,4\}\}$ | 1 —— 3 <br> 2 —— 4 | 1 —— 3 <br> 2 —— 4 |
| | $\{\{1,4\},\{2,3\}\}$ | 1 ✕ 3 <br> 2 ✕ 4 | 1 ✕ 3 <br> 2 ✕ 4 |
| | $\{\{1,2\},\{3,4\}\}$ | 1 ⌒ 3 <br> 2 ⌣ 4 | 1 ⌒ 3 <br> 2 Ω⌣Ω 4 |

Figure 4: Spanning sets for the first and second order commutants of the orthogonal and (unitary) symplectic groups; for both groups the elements are indexed respectively by the set of pairs of the sets of two and four elements [39–41]. Here we highlight the connection between this abstract presentation and the concrete realisation afforded by the tensor-network notation. The operator $\Omega$ that appears in the symplectic case is the canonical symplectic form.

symmetric group, which one could embed within $\mathfrak{B}_k$ by dropping the linear combinations and considering only the pairings such that, for some ordering of the elements, each pair contains an element from both the first and last $k$ elements [40]).

Explicitly, $\sigma = \{\{\lambda_1, \sigma(\lambda_1)\}, \dots, \{\lambda_k, \sigma(\lambda_k)\}\} \in \mathfrak{B}_k$ is represented by

$$F_d(\sigma) = \sum_{i_1,\dots,i_{2k}=0}^{d-1} |i_{k+1},\dots,i_{2k}\rangle\langle i_1,\dots,i_k| \prod_{\gamma=1}^{k} \delta_{i_{\lambda_\gamma}, i_{\sigma(\lambda_\gamma)}}. \tag{55}$$

From the above equation we can readily obtain a spanning set (of size three) of the second-order commutant (i.e. by setting $k = 2$ and going through the possible $\sigma \in \mathfrak{B}_2$). This is depicted in Fig. 4, where we see that the pairs that constitute an element $\sigma \in \mathfrak{B}_k$ have in the quantum computation context an elegant interpretation as the endpoints of wires in the graphical notation [26]. Multiplication in the algebra then corresponds to the concatenation of diagrams, with the possibility of the formation of "loops" being accounted for by the indeterminant $d$ (in our case $d = 2^n$, corresponding to the trace of the identity operator); in other words, multiplying two diagrams yields another valid diagram, as well as a non-negative power of $d$. From either Fig. 4 or Eq. (55) directly we can see that the three quadratic symmetries produced by this characterisation are $\mathbb{1}_{\mathcal{H}^{\otimes 2}}$, $\mathbb{S}$ and $|\Phi\rangle\langle\Phi|$, where $|\Phi\rangle = \sum_i |ii\rangle$ is an (unnormalised) maximally mixed state across the two copies of $\mathcal{H}$. By instead setting $k = 1$ we discover that there is a single linear symmetry, the identity.[8]

Amongst other nice consequences, the fact that $\mathbb{O}(d)$ possesses one linear and three quadratic symmetries immediately implies by Prop. 1 that the commutator graph of an orthogonal Pauli DLA has three connected components, and that each of them furnishes an irreducible representation of $\mathbb{O}(d)$. In fact, it is straightforward to explicitly identify what these components must be. One of them, of course, contains only the identity Pauli string, and the other two follow from noticing that each Pauli string is either symmetric or anti-symmetric, and that (anti-)symmetry is preserved by the action of $\mathbb{O}(d)$. Denoting these components as

---

[8]This can also be seen by noticing that the action of $\mathbb{O}$ on $\mathcal{H}$ is irreducible, and appealing to Schur's lemma.

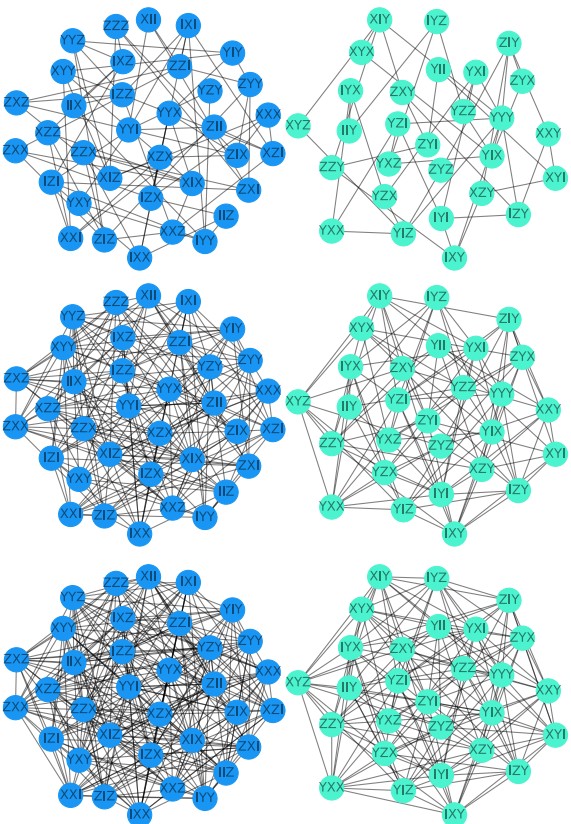

Figure 5: The (non-trivial components) of the commutator graphs in the case of $\mathfrak{a}_{16}(n) \cong \mathfrak{so}(2^n)$, where for the graphs of the top, middle and bottom rows we take a generating set given respectively by $\{XY, YX, YZ, ZY\}$, the proceeding set augmented with its first order commutators, and the Lie closure of the set.

$I_d$, $S_d$ and $A_d$, we can then simply count to find that they contain 1, $d(d+1)/2 - 1$ and $d(d-1)/2$ vertices respectively. From Eq. (8) we therefore obtain a second spanning set for $\text{comm}(\mathbb{O}(d), 2)$, namely $\{\mathbb{1}, \sum_{S \in S_d} S \otimes S, \sum_{A \in A_d} A \otimes A\}$; one can readily verify the consistency of this result with the standard spanning set of Fig. 4.

Across Figs. 5 and 6 we explore some differences introduced by considering different generating sets of the shared Lie algebra

$$\langle \{X_i Y_{i+1}, Y_i X_{i+1}, Y_i Z_{i+1}, Z_i Y_{i+1}\}_{i=1}^{n-1} \rangle_{\text{Lie}} \cong \mathfrak{so}(2^n). \tag{56}$$

In Fig. 5 we depict the $n = 3$ commutator graph (excluding the identity component) for increasingly large choices of generating set; as expected, the vertices of the connected components do not change, but additional edges within components emerge. In Fig. 6 we plot the increase in graph complexity as a function of time for initial strings Heisenberg-evolved under Hamiltonians corresponding to the chosen generating set. We find that larger generating sets generically lead to more complicated dynamics and a more rapid initial rise in complexity, but also lower long-time graph complexity due to the shorter paths enabled by the increased connectivity.

**Symplectic evolution.** The case of evolution by random symplectic unitaries bears a considerable resemblance to the orthogonal case, which as we will see stems from the close representation-theoretic descriptions of their commutants. We begin by recalling that the group $\mathbb{SP}(d/2)$ of symplectic unitaries consists of the $d \times d$ unitary matrices that satisfy

$$U^T \Omega U = \Omega, \tag{57}$$

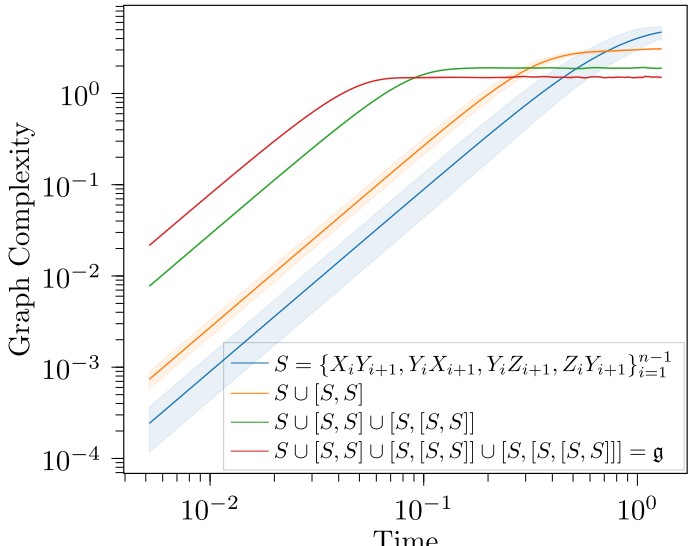

Figure 6: The graph complexity of a Heisenberg-evolved Pauli string ($YXXXX$) under random orthogonal dynamics. In each case, a Hamiltonian is sampled by uniformly randomly weighting the elements of the generating set; the mean and standard deviation over five runs is displayed (note that the shaded region corresponding to the standard deviation is very thin for the larger generating sets). A larger generating set allows for more rapid initial growth in complexity, but ultimately stabilises at a lower value due to the higher connectivity and therefore shorter path lengths within the graph. For short times we observe the universal $t^2$ scaling predicted by Lemma 10.

where $\Omega$ is a non-degenerate anti-symmetric bilinear form satisfying

$$\Omega^2 = -\mathbb{1}_d, \qquad \Omega\Omega^T = \mathbb{1}_d. \tag{58}$$

When necessary, we will choose the canonical representation $\Omega = iY \otimes I^{\otimes(n-1)}$. We will consider the $n$-qubit Pauli Lie algebra isomorphic to $\mathfrak{sp}(2^{n-1})$ generated by [41]

$$\{Y_i\}_{i=1}^n \cup \{X_iY_{i+1}, \ Y_iX_{i+1}\}_{i=1}^{n-1} \cup X_1 \cup Z_1Z_2. \tag{59}$$

We note the lack of translational invariance of the set of generators, induced by our specific choice of the symplectic form $\Omega$ [41]. The structure of the $k^{\text{th}}$ order commutant is captured by a representation $G_d$ of the Brauer algebra $\mathfrak{B}_k(-d)$, explicitly given by [41]

$$
\begin{aligned}
G_d(\sigma) = \sum_{i_1,\ldots,i_{2k}=1}^{d} \prod_{\gamma=1}^{t} & \Omega_{\sigma(\lambda_\gamma)}^{h(\lambda_\gamma,\sigma(\lambda_\gamma))} |i_{k+1}, i_{k+2}, \ldots, i_{2k}\rangle \\
& \times \langle i_1, i_2, \ldots, i_k| \ \Omega_{\sigma(\lambda_\gamma)}^{h(\lambda_\gamma,\sigma(\lambda_\gamma))} \delta_{i_{\lambda_\gamma}, i_{\sigma(\lambda_\gamma)}},
\end{aligned}
\tag{60}
$$

where one notes the similarity to Eq. (55). Here $h(\lambda_\gamma, \sigma(\lambda_\gamma)) = 1$ if $\lambda_\gamma, \sigma(\lambda_\gamma) \leq n$ or $\lambda_\gamma, \sigma(\lambda_\gamma) > n$, and is zero otherwise. As in the previous example, this somewhat formidable-looking expression turns out to yield easily interpretable operators, which we depict graphically in Fig. 4. We see that again there exist one linear[9] and three quadratic symmetries. From the known simplicity of $\mathfrak{sp}$ [35] we see that one of these components will correspond to the

---

[9]Alternatively, by Schur's lemma combined with the irreducibility (indeed, transitivity [45,46]) of the action of $\mathbb{SP}$ on $\mathcal{H}$.

adjoint module of $\mathfrak{sp}(2^{n-1})$, with a number of vertices given by $\dim \mathfrak{sp}(2^{n-1}) = 2^{n-1}(2^n+1)$; it follows that the other two necessarily possess 1 and $2^{n-1}(2^n-1)-1$ vertices. So the situation is indeed quite similar to the orthogonal case.

Interestingly, we can leverage the recent result [47] that Haar random symplectic states form a unitary state $t$-design for all $t \geq 1$, i.e. for all $|\psi\rangle \in \mathcal{H}$,

$$\int_{U \sim \mu_{\mathbb{U}(d)}} U^{\dagger \otimes t} |\psi\rangle\langle\psi|^{\otimes t} U^{\otimes t} = \int_{U \sim \mu_{\mathbb{SP}(d/2)}} U^{\dagger \otimes t} |\psi\rangle\langle\psi|^{\otimes t} U^{\otimes t}, \tag{61}$$

to deduce properties of the vertex placement in the commutator graph. The above equation implies that the distributions of unitary and symplectic (pure) states are *indistinguishable* [47], from which we conclude that the (traceless) Pauli strings comprised of only $I$ and $Z$ cannot all appear in the same component. To see this we note that, were it not the case, one could form (for example) the pure state

$$|0\rangle\langle 0|^{\otimes n} = 2^{-n} \sum_{P \in \{I,Z\}^{\otimes n}} P, \tag{62}$$

which upon Heisenberg evolution by random symplectic unitaries would behave noticeably differently to the unitary case (by e.g. always having zero overlap with Pauli strings from the third component), contradicting their indistinguishability. The orthogonal group, on the other hand, has no such restriction (failing as it does to form a unitary state 2-design for certain reference states [43, 44]), and indeed we find that the (traceless) Pauli strings comprised of only $I$ and $Z$ all appear in the same component of the commutator graph for the example of the orthogonal DLA of Fig. 5.

# 5 Discussion

The various computations of this paper go "beyond" the dynamical Lie algebras of ensembles of systems in several ways. Firstly, and in the sense of the word used by Ref. [6], the commutator graph describes the Heisenberg evolution of *any* operator on the Hilbert space of the system, not just those within the DLA, themselves described by the adjoint module (i.e., a *subset* of the connected components of the commutator graph). In this sense one obtains a far more complete picture of the dynamics, as observables of interest need not lie within the DLA. One immediate consequence, for example, is that the cost of exactly classically simulating the Heisenberg evolution of a given operator under the dynamics is upper bounded by the size of the connected component within which it resides.

The graph picture also adapts nicely to the approximate simulation technique of truncated Pauli propagation [48–53], wherein one Heisenberg-evolves an initial operator $O$ through a circuit $U$ while, at each timestep, dropping any contributions to $O$ from Pauli strings of weight above some constant $k$. From the commutator graph perspective, this corresponds to simulation as usual, modulo the deletion of vertices corresponding to Pauli strings of weight above $k$. Generically, this will reduce the size of the connected component of a given string, easing its simulation. It would be interesting to investigate the induced breaking-up of components as a function of $k$ for various models; graph-theoretically, this is captured by the notion of *vertex-connectivity*.

Secondly, as we have discussed, the commutator graph captures information that is more fine-grained than the DLA. Intuitively, the information "missed" by the DLA corresponds to short-time phenomena that is invisible once one forgets the distinguished set of (usually local) operators that generate the dynamics. At the graph level, this manifests as identical DLAs with different generating sets leading to graphs with vertices that form the same connected

components, but with the components having differing internal edge structures. It is then unsurprising that results which involve averaging over the entire ensemble (without making reference to the generating set) depend only on properties of the graph that are invariant to changing the internal edges of connected components (without, that is, breaking up the component). Proposition 1, for example, shows that the frame potential is sensitive only to the number of isolated vertices and the number of connected components, which are both certainly invariant under such changes.

Finally, we note that throughout this work we have focused on average-case statements at the level of entire families of Hamiltonians, coupled with the guarantee of Prop. 6 of the typicality of the average case. In a somewhat different direction, one can imagine employing commutator graph based techniques to analyse individual instances of such a family; indeed, one is often particularly interested in specific Hamiltonians demonstrating somehow atypical behaviour. For example, the quantum Ising spin chain exhibits a quantum phase transition for specific values of its parameters. Occurring as they may in a vanishing fraction of the full manifold of dynamics, such atypical instances can be invisible to average-case analysis. For example, and as mentioned above, the commutator graph arises naturally in the problem of simulating a Heisenberg-evolving Pauli string. In the specific-instance context one is led immediately to the idea of *weighted* commutator graphs, the study of which we leave for future work.

## Acknowledgments

**Funding information**  MW acknowledges the support of the Australian Government Research Training Program Scholarship. ND acknowledges funding by the Deutsche Forschungsgemeinschaft (DFG, German Research Foundation) under Germany's Excellence Strategy - Cluster of Excellence Matter and Light for Quantum Computing (ML4Q) EXC 2004/1 - 390534769. MU and MW acknowledge funding from the Australian Army Research through Quantum Technology Challenge program. Computational resources were provided by the Pawsey Supercomputing Research Center through the National Computational Merit Allocation Scheme (NCMAS). KM acknowledges the support of the Australian Research Council's Discovery Projects DP210100597 and DP220101793.

## A  Proofs

**Proposition 1.** *If both the DLA $\mathfrak{g}$ of an ensemble of unitaries $G = \exp \mathfrak{g}$ and its linear symmetries have bases consisting of Pauli strings, then the frame potential $F_G^{(2)}$ can be read off the commutator graph as:*

$$F_G^{(2)} = \#(\text{isolated vertices}) \cdot \#(\text{components}). \tag{26}$$

*Proof.* We present two perspectives on this result. Firstly, as discussed in Section 2.2 (see also e.g. [26]), the frame potential $F_G^{(2)}$ is given by the number of linearly independent second order symmetries of the dynamics (i.e. $Q \in \mathcal{L}^{\otimes 2} : [Q, U^{\otimes 2}] = 0 \; \forall U \in G$), so by Equation (8) we have $F_G^{(2)} = |J| \times \#(\text{components})$, with $|J|$ the number of (linear) symmetries. As these symmetries are precisely the single-element components of the graph, the result follows.

Alternately, and as a warm-up exercise in the sort of Pauli string manipulations we will be doing later, we can employ the relation [28]

$$F_G^{(2)} = \frac{1}{d^4} \sum_{A_1, B_1, A_2, B_2 \in \mathbb{P}} \left| \int_G d\mu_G(U) \, \text{tr}\big[A_1 U B_1 U^\dagger A_2 U B_2 U^\dagger\big] \right|^2, \tag{A.1}$$

where $\mathbb{P}$ is the set of Pauli strings, and directly compute $F_G^{(2)}$. To start, we have

$$F_G^{(2)} = \frac{1}{d^4} \sum_{A_1,B_1,A_2,B_2 \in \mathbb{P}} \left| \int_G d\mu_G(U)\, \mathrm{tr}\big[A_1 U B_1 U^\dagger A_2 U B_2 U^\dagger\big] \right|^2 \tag{A.2}$$

$$= \frac{1}{d^4} \sum_{A_1,B_1,A_2,B_2 \in \mathbb{P}} \left| \int_G d\mu_G(U)\, \mathrm{tr}\Big[(A_1 \otimes A_2)\big(U^{\otimes 2}(B_1 \otimes B_2)(U^\dagger)^{\otimes 2}\big)\mathbb{S}\Big] \right|^2 \tag{A.3}$$

$$= \frac{1}{d^4} \sum_{A_1,B_1,A_2,B_2 \in \mathbb{P}} \left| \mathrm{tr}\left[ (A_1 \otimes A_2)\Big(\sum_{j,\kappa} Q_{j,\kappa} \frac{\langle Q_{j,\kappa}, B_1 \otimes B_2 \rangle_{\mathrm{HS}}}{\langle Q_{j,\kappa}, Q_{j,\kappa} \rangle_{\mathrm{HS}}}\Big)\mathbb{S} \right] \right|^2 \tag{A.4}$$

$$= \frac{1}{d^4} \sum_{A_1,B_1,A_2,B_2 \in \mathbb{P}} \left| \mathrm{tr}\left[ (A_1 \otimes A_2)\Big(\sum_{j,\kappa} \frac{Q_{j,\kappa}}{|C_\kappa| d^2} \Big\langle \sum_{S \in C_\kappa} S \otimes L_j S, B_1 \otimes B_2 \Big\rangle_{\mathrm{HS}}\Big)\mathbb{S} \right] \right|^2, \tag{A.5}$$

where in Eq. (A.3) we have used the "swap trick" [26], $\mathrm{tr}\,AB = \mathrm{tr}[(A \otimes B)\mathbb{S}]$, with $\mathbb{S}$ the swap operator, in Eq. (A.4) the fact that $\int_G d\mu(U)\, U^{\dagger \otimes 2}(-)U^{\otimes 2}$ projects onto the quadratic symmetries of $G$, and in Eq. (A.5) the basis Eq. (8) of the quadratic symmetries. This is the part that requires $\mathfrak{g}$ to be a Pauli string DLA with a Pauli string basis of its linear symmetries, as otherwise it is not guaranteed that Eq. (8) gives the required basis. We have also used

$$\langle Q_{j,\kappa}, Q_{j,\kappa} \rangle_{\mathrm{HS}} = \left\langle \sum_{S \in C_\kappa} S \otimes L_j S, \sum_{T \in C_\kappa} T \otimes L_j T \right\rangle_{\mathrm{HS}}$$
$$= |C_\kappa| d^2,$$

as for Pauli strings $S$ and $T$ we have $\mathrm{tr}\,ST = d\delta_{S,T}$. Continuing on for a little, we have:

$$F_G^{(2)} = \frac{1}{d^8} \sum_{A_1,B_1,A_2,B_2 \in \mathbb{P}} \left| \mathrm{tr}\left[ (A_1 \otimes A_2)\Big(\sum_{j,\kappa} \frac{Q_{j,\kappa}}{|C_\kappa|} \sum_{S \in C_\kappa} \mathrm{tr}\big[S^\dagger B_1\big]\mathrm{tr}\big[(L_j S)^\dagger B_2\big]\Big)\mathbb{S} \right] \right|^2 \tag{A.6}$$

$$= \frac{1}{d^8} \sum_{A_1,B_1,A_2,B_2 \in \mathbb{P}} \left| \sum_{j,\kappa} \frac{1}{|C_\kappa|} \sum_{S,T \in C_\kappa} d\delta_{S,B_1}\, \mathrm{tr}\big[S L_j B_2\big] \mathrm{tr}\big[A_1 T A_2 L_j T\big] \right|^2 \tag{A.7}$$

$$= \frac{1}{d^6} \sum_{A_1,B_1,A_2,B_2 \in \mathbb{P}} \left| \sum_j \frac{1}{|C_{B_1}|} \sum_{T \in C_{B_1}} \mathrm{tr}\big[B_1 L_j B_2\big] \mathrm{tr}\big[A_1 T A_2 L_j T\big] \right|^2 \tag{A.8}$$

$$= \frac{1}{d^6} \sum_{A_1,B_1,A_2,B_2 \in \mathbb{P}} \frac{1}{|C_{B_1}|^2} \sum_{j,j'} \sum_{T,T' \in C_{B_1}} \mathrm{tr}\big[B_1 L_j B_2\big] \mathrm{tr}\big[B_1 L_{j'} B_2\big]^* \mathrm{tr}\big[A_1 T A_2 L_j T\big]$$
$$\times \mathrm{tr}\big[A_1 T' A_2 L_{j'} T'\big]^* \tag{A.9}$$

$$= \frac{1}{d^6} \sum_{A_1,B_1,A_2,B_2 \in \mathbb{P}} \frac{1}{|C_{B_1}|^2} \sum_j \sum_{T,T' \in C_{B_1}} \mathrm{tr}\big[B_1 L_j B_2\big] \mathrm{tr}\big[B_2 L_j B_1\big] \mathrm{tr}\big[A_1 T A_2 L_j T\big]$$
$$\times \mathrm{tr}\big[T' L_j A_2 T' A_1\big]. \tag{A.10}$$

Where in Eq. (A.7) we have again used the orthogonality of the Paulis with respect to the Hilbert-Schmidt inner product, and in Eq. (A.11) the self-adjointness of the Paulis combined with the fact that $\mathrm{tr}[M] = \mathrm{tr}\big[M^T\big]\ \forall M \in \mathrm{End}(\mathcal{H})$. We have also used that $\mathrm{tr}\big[B_1 L_j B_2\big]$ and

$\mathrm{tr}\big[B_1 L_{j'} B_2\big]^*$ can only be simultaneously non-vanishing if $j = j'$. Next up we use the relation [26] $(1/d)\sum_{P\in\mathbb{P}} P \otimes P = \mathbb{S}$ to conclude that

$$\sum_{B_2\in\mathbb{P}} \mathrm{tr}\big[B_1 L_j B_2\big]\mathrm{tr}\big[B_2 L_j B_1\big] = \sum_{B_2\in\mathbb{P}} \mathrm{tr}\big[(B_1\otimes L_j)(L_j\otimes B_1)(B_2\otimes B_2)\big] \tag{A.11}$$

$$= d\,\mathrm{tr}\big[(B_1\otimes L_j)(L_j\otimes B_1)\mathbb{S}\big] \tag{A.12}$$

$$= d\,\mathrm{tr}\big[B_1 L_j L_j B_1\big] \tag{A.13}$$

$$= d^2\,, \tag{A.14}$$

and, similarly,

$$\sum_{A_1,A_2\in\mathbb{P}} \mathrm{tr}\big[A_1 T A_2 L_j T\big]\mathrm{tr}\big[T' L_j A_2 T' A_1\big] = \sum_{A_1,A_2\in\mathbb{P}} \mathrm{tr}\big[(A_1\otimes A_1)(T\otimes T')(\mathbb{1}\otimes L_j) \tag{A.15}$$

$$\times (A_2\otimes A_2)(L_j\otimes\mathbb{1})(T\otimes T')\big]$$

$$= d^2\,\mathrm{tr}\big[\mathbb{S}(T\otimes T')(\mathbb{1}\otimes L_j)\mathbb{S}(L_j\otimes\mathbb{1})(T\otimes T')\big] \tag{A.16}$$

$$= d^2\,\mathrm{tr}\big[T T'\big]\mathrm{tr}\big[T' L_j^2 T\big] \tag{A.17}$$

$$= d^4 \delta_{T,T'}\,. \tag{A.18}$$

Putting it all together, we have:

$$F_G^{(2)} = \sum_{B_1\in\mathbb{P}} \frac{1}{|C_{B_1}|^2} \sum_j \sum_{T,T'\in C_{B_1}} \delta_{T,T'} \tag{A.19}$$

$$= |J| \sum_{B_1\in\mathbb{P}} \frac{1}{|C_{B_1}|} \tag{A.20}$$

$$= |J| \sum_\kappa 1 \tag{A.21}$$

$$= |J| \cdot \#(\text{connected components}), \tag{A.22}$$

which is what we wanted to show. In the last few lines we have broken the sum over all Paulis into a sum over sums over Paulis from each component, i.e.

$$\sum_{B_1\in\mathbb{P}} (1/|C_{B_1}|) = \sum_\kappa \sum_{B_1\in C_\kappa} (1/|C_{B_1}|) = \sum_\kappa 1\,.$$

$\square$

**Proposition 2.** *Let $P,Q,R,S$ be Pauli strings belonging respectively to the connected components $C_P, C_Q, C_R, C_S$ of the graph. Then*

$$\underset{U\sim\mu_G}{\mathbb{E}} \mathrm{tr}\big[PUQU^\dagger RUSU^\dagger\big] = d^{-1}\langle\!\langle RP|\mathsf{T}_G^{(1)}|QS\rangle\!\rangle \left(1 - \frac{2|\{T\in C_Q : \{P,T\}=0\}|}{|C_Q|}\right). \tag{28}$$

*This expression is zero unless there exists a linear symmetry $L$ such that $RP, QS \propto L$.*

*Proof.* As in Props. 1, 3 and 7 we employ the swap trick and Weingarten calculus, utilising the

basis Eq. (8) of the quadratic symmetries:

$$\mathop{\mathbb{E}}_{U\sim\mu_G}\mathrm{tr}\left[PUQU^\dagger RUSU^\dagger\right]=\mathop{\mathbb{E}}_{U\sim\mu_G}\mathrm{tr}\left[(P\otimes R)U^{\otimes2}(Q\otimes S)U^{\dagger\otimes2}\mathbb{S}\right] \tag{A.23}$$

$$=\mathrm{tr}\left[(P\otimes R)\Big(\sum_{j,\kappa}Q_{j,\kappa}\frac{\langle Q_{j,\kappa},Q\otimes S\rangle_{\mathrm{HS}}}{\langle Q_{j,\kappa},Q_{j,\kappa}\rangle_{\mathrm{HS}}}\Big)\mathbb{S}\right] \tag{A.24}$$

$$=\mathrm{tr}\left[(P\otimes R)\Big(\sum_{j,\kappa}\frac{Q_{j,\kappa}}{|C_\kappa|d^2}\Big\langle\sum_{T\in C_\kappa}T\otimes L_jT,Q\otimes S\Big\rangle_{\mathrm{HS}}\Big)\mathbb{S}\right] \tag{A.25}$$

$$=\frac{1}{d^2}\mathrm{tr}\left[(P\otimes R)\Big(\sum_{j,\kappa}\frac{Q_{j,\kappa}}{|C_\kappa|}\sum_{T\in C_\kappa}\mathrm{tr}[TQ]\,\mathrm{tr}[L_jTS]\Big)\mathbb{S}\right]. \tag{A.26}$$

But $\mathrm{tr}[TQ]=d\delta_{T,Q}$, so we set $T=Q$ and the sum over $\kappa$ collapses to the component of $Q$. We carry on:

$$\frac{1}{d^2}\mathrm{tr}\left[(P\otimes R)\Big(\sum_{j,\kappa}\frac{Q_{j,\kappa}}{|C_\kappa|}\sum_{T\in C_\kappa}\mathrm{tr}[TQ]\,\mathrm{tr}[L_jTS]\Big)\mathbb{S}\right]=\frac{1}{d}\mathrm{tr}\left[(P\otimes R)\Big(\sum_j\frac{Q_{j,C_Q}}{|C_Q|}\mathrm{tr}[L_jQS]\Big)\mathbb{S}\right] \tag{A.27}$$

$$=\sum_j\frac{\mathrm{tr}[L_jQS]}{d|C_Q|} \tag{A.28}$$

$$\times\mathrm{tr}\left[(P\otimes R)\Big(\sum_{T\in C_Q}T\otimes L_jT\Big)\mathbb{S}\right]$$

$$=\sum_j\sum_{T\in C_Q}\frac{\mathrm{tr}[L_jQS]}{d|C_Q|}\mathrm{tr}[PTRL_jT]. \tag{A.29}$$

Now, by the fact that Pauli strings always commute or anticommute (and regardless square to the identity) we have $\mathrm{tr}[PTRL_jT]=\pm\mathrm{tr}[L_jPR]$, and so

$$\sum_j\frac{\mathrm{tr}[L_jQS]}{d|C_Q|}\sum_{T\in C_Q}\mathrm{tr}[PTRL_jT]=\sum_j\frac{\mathrm{tr}[L_jQS]\,\mathrm{tr}[PRL_j]}{d|C_Q|}\sum_{T\in C_Q}\begin{cases}1,&\text{if }[P,T]=0,\\-1,&\text{otherwise}\end{cases} \tag{A.30}$$

$$=\sum_j\frac{\mathrm{tr}[L_jQS]\,\mathrm{tr}[PRL_j]}{d}\left(1-\frac{2|\{T\in C_Q:\{P,T\}=0\}|}{|C_Q|}\right). \tag{A.31}$$

The next step is perhaps most easily seen in the vectorised notation: recalling $\langle\!\langle A|B\rangle\!\rangle=\mathrm{tr}[A^\dagger B]$ and Eq. (12) we have

$$\sum_j\mathrm{tr}[L_jQS]\,\mathrm{tr}[PRL_j]=\sum_j\langle\!\langle RP|L_j\rangle\!\rangle\langle\!\langle L_j|QS\rangle\!\rangle=\langle\!\langle RP|\Big(\sum_j|L_j\rangle\!\rangle\langle\!\langle L_j|\Big)|QS\rangle\!\rangle=\langle\!\langle RP|\mathsf{T}_G^{(1)}|QS\rangle\!\rangle, \tag{A.32}$$

and conclude

$$\mathop{\mathbb{E}}_{U\sim\mu_G}\mathrm{tr}\left[PUQU^\dagger RUSU^\dagger\right]=d^{-1}\langle\!\langle RP|\mathsf{T}_G^{(1)}|QS\rangle\!\rangle\left(1-\frac{2|\{T\in C_Q:\{P,T\}=0\}|}{|C_Q|}\right). \tag{A.33}$$

Finally, from the left-most of the expressions in Eq. (A.32) it is clear that the overall expression Eq. (A.33) will be zero unless one of the Paulis $L_j$ is proportional to both $RP$ and $QS$. $\qquad\square$

**Corollary 3.** *Pick two Pauli strings $V$ and $W$, belonging respectively to the connected components $C(V)$ and $C(W)$ of the graph. Then we have*

$$\mathop{\mathbb{E}}_{U \sim \mu_G} F(W, U^\dagger V U) = 1 - \frac{2|\{T \in C(V) : \{W, T\} = 0\}|}{|C(V)|} \tag{31}$$

$$= 1 - \frac{2|\{T \in C(W) : \{V, T\} = 0\}|}{|C(W)|} . \tag{32}$$

*Proof.* From the definition of the OTOC and the result of Prop. 2 we have

$$\mathop{\mathbb{E}}_{U \sim \mu_G} F(W, U^\dagger V U) = \frac{1}{d} \mathop{\mathbb{E}}_{U \sim \mu_G} \mathrm{tr}\left[W U^\dagger V U W U^\dagger V U\right] \tag{A.34}$$

$$= d^{-2} \langle\!\langle W^2 | \mathsf{T}_G^{(1)} | W^2 \rangle\!\rangle \left(1 - \frac{2|\{T \in C(V) : \{W, T\} = 0\}|}{|C(V)|}\right) \tag{A.35}$$

$$= \frac{\mathrm{tr}[\mathbb{1}]^2}{d^2} \left(1 - \frac{2|\{T \in C(V) : \{W, T\} = 0\}|}{|C(V)|}\right) \tag{A.36}$$

$$= 1 - \frac{2|\{T \in C(V) : \{W, T\} = 0\}|}{|C(V)|} . \tag{A.37}$$

The statement with the roles of $V$ and $W$ reversed follows similarly; alternately we have

$$\mathop{\mathbb{E}}_{U \sim \mu_G} F(W, U^\dagger V U) = \frac{1}{d} \mathop{\mathbb{E}}_{U \sim \mu_G} \mathrm{tr}\left[W U^\dagger V U W U^\dagger V U\right] \tag{A.38}$$

$$= \frac{1}{d} \mathop{\mathbb{E}}_{U \sim \mu_G} \mathrm{tr}\left[V U^\dagger W U V U^\dagger W U\right] \tag{A.39}$$

$$= \mathop{\mathbb{E}}_{U \sim \mu_G} F(V, U^\dagger W U) \tag{A.40}$$

$$= 1 - \frac{2|\{T \in C(W) : \{V, T\} = 0\}|}{|C(W)|} , \tag{A.41}$$

using the previous result, the cyclicity of the trace, and the invariance of the Haar measure under the transformation $U \mapsto U^\dagger$. $\qquad\square$

**Corollary 4.** *Pick two Pauli strings $V$ and $W$, belonging respectively to the connected components $C(V)$ and $C(W)$ of the graph. We have:*

$$\frac{|\{T \in C(W) : [V, T] = 0\}|}{|C(W)|} = \frac{|\{T \in C(V) : [W, T] = 0\}|}{|C(V)|} , \tag{33}$$

*and*

$$\frac{|\{T \in C(W) : \{V, T\} = 0\}|}{|C(W)|} = \frac{|\{T \in C(V) : \{W, T\} = 0\}|}{|C(V)|} , \tag{34}$$

*where $|S|$ denotes the number of elements of the set $S$.*

*Proof.* From Corollary 3 we have

$$\frac{|\{T \in C(W) : \{V, T\} = 0\}|}{|C(W)|} = \frac{|\{T \in C(V) : \{W, T\} = 0\}|}{|C(V)|} .$$

As Pauli strings either commute or anticommute we have $\forall W$

$$|C(V)| = |\{T \in C(V) : \{W, T\} = 0\}| + |\{T \in C(V) : [W, T] = 0\}| ,$$

and similarly for $V \leftrightarrow W$. Combining this with the previous result immediately implies

$$\frac{|\{T \in C(W) : [V, T] = 0\}|}{|C(W)|} = \frac{|\{T \in C(V) : [W, T] = 0\}|}{|C(V)|} .$$

$\qquad\square$

**Corollary 5.** *If $\mathfrak{g} = \mathrm{span}_{i\mathbb{R}}(\mathcal{G})$ is a simple Lie algebra, then all nodes in the frustration graph of any Hamiltonian of the form Eq. (1) have the same degree.*

*Proof.* The *commutator* graph has a connected component for each ideal of the DLA (among other components). Here the DLA is equal to $\mathfrak{g}$, as by assumption the Hamiltonian terms already form a (simple) Lie algebra. As the DLA is simple, any strings $P_i, P_j \in \mathcal{G}$ are in the same component $C(P_i)$ of the graph, which consists of and only of the strings in $\mathcal{G}$. Now we can use Corollary 3:

$$\mathop{\mathbb{E}}_{U \sim \mu_G} F(P_i, U^\dagger P_j U) = 1 - \frac{2}{|C(P_i)|} |\{T \in C(P_i) : \{P_j, T\} = 0\}| = 1 - \frac{2 \deg(P_i)}{\dim \mathfrak{g}}, \tag{A.42}$$

and by symmetry $\deg(P_i) = \deg(P_j)$. By considering the analogous expression for each pair of nodes in $C(P_i)$ we conclude that all the vertices in the *frustration* graph have the same degree. $\qquad\square$

**Proposition 6.** *The averages of Proposition 2 and Corollaries 3 and 5 are typical for a classical reductive DLA $\mathfrak{g} = \bigoplus_i \mathfrak{g}_i$ with $\mathfrak{g}_i \in \{\mathfrak{so}(k_i), \mathfrak{su}(k_i), \mathfrak{u}(k_i), \mathfrak{sp}(2k_i)\}$ in the sense that the probability that a given single-shot evaluation with a $U \sim \mu_G$ deviates from the calculated averages is exponentially suppressed,*

$$\mathbb{P}_{U \sim \mu_G}(|f(U) - \mathbb{E}[f]| \geq \epsilon) \leq 2e^{-(k-2)^2 \epsilon^2/384}, \tag{35}$$

*with $k = \min\{k_i\}_i$, and $f(U) = \mathrm{tr}\left[PUQU^\dagger RUSU^\dagger\right]$, where $P, Q, R$ and $S$ are Pauli strings.*

We note that our proof is inspired by, and bears considerable resemblance to, proofs of similar statements found in Refs. [17, 54].

*Proof.* First, let us say that a function $f : G \to \mathbb{R}$ is *L-Lipschitz continuous*, if, for all $U, V \in G$,

$$\left|f(U) - f(V)\right| \leq L\|U - V\|_2, \tag{A.43}$$

where $\|T\|_2 = \sqrt{\mathrm{tr}[T^\dagger T]}$ is the Hilbert-Schmidt norm of $T \in \mathrm{End}(\mathcal{H})$. The main technical result we will need is that by Thm. 5.17 of Ref. [55] we have concentration of measure on $G$; specifically, if $f : G \to \mathbb{R}$ is $L$-Lipschitz continuous, then the probability of $f$ deviating from its average by more than $\epsilon$ is bounded as

$$\mathbb{P}_{U \sim \mu_G}(|f(U) - \mathbb{E}f| \geq \epsilon) \leq 2e^{-(k-2)^2 \epsilon^2/24L^2}, \tag{A.44}$$

where $k = \min\{k_i\}_i$. So the result will follow if we can show that $f : G \to \mathbb{R}$, $f(T) = \mathrm{tr}\left[ATBT^\dagger CTDT^\dagger\right]$ is $L$-Lipschitz continuous with $L = 4$. We have:

$$|f(S) - f(T)| = \frac{1}{d}\left|\mathrm{tr}\left[ASBS^\dagger CSDS^\dagger\right] - \mathrm{tr}\left[ATBT^\dagger CTDT^\dagger\right]\right| \tag{A.45}$$

$$= \frac{1}{d}\left|\mathrm{tr}\left[\mathbb{S}\left((A \otimes C)S^{\otimes 2}(B \otimes D)S^{\dagger \otimes 2} - (A \otimes C)T^{\otimes 2}(B \otimes D)T^{\dagger \otimes 2}\right)\right]\right| \tag{A.46}$$

$$\leq \frac{1}{d}\left\|S^{\otimes 2}(B \otimes D)S^{\dagger \otimes 2} - T^{\otimes 2}(B \otimes D)T^{\dagger \otimes 2}\right\|_1 \cdot \left\|(A \otimes C)\mathbb{S}\right\|_\infty \tag{A.47}$$

$$= \frac{1}{d}\left\|S^{\otimes 2}(B \otimes D)S^{\dagger \otimes 2} - T^{\otimes 2}(B \otimes D)T^{\dagger \otimes 2}\right\|_1, \tag{A.48}$$

where in Eq. (A.46) we used the swap trick, in Eq. (A.47) the matrix Hölder inequality $|\operatorname{tr} AB| \leq \|A\|_1 \cdot \|B\|_\infty$, and in Eq. (A.48) that unitary operators have $\infty$-norm 1. Continuing on, we have (strategically adding zero)

$$
\begin{aligned}
\left| f(S) - f(T) \right| &= \frac{1}{d} \left\| S^{\otimes 2}(B \otimes D)S^{\dagger \otimes 2} - S^{\otimes 2}(B \otimes D)T^{\dagger \otimes 2} + S^{\otimes 2}(B \otimes D)T^{\dagger \otimes 2} \right. \\
&\qquad \left. - T^{\otimes 2}(B \otimes D)T^{\dagger \otimes 2} \right\|_1 
\end{aligned}
\tag{A.49}
$$

$$
\begin{aligned}
&\leq \frac{1}{d} \left\| S^{\otimes 2}(B \otimes D)S^{\dagger \otimes 2} - S^{\otimes 2}(B \otimes D)T^{\dagger \otimes 2} \right\|_1 \\
&\quad + \frac{1}{d} \left\| S^{\otimes 2}(B \otimes D)T^{\dagger \otimes 2} - T^{\otimes 2}(B \otimes D)T^{\dagger \otimes 2} \right\|_1
\end{aligned}
\tag{A.50}
$$

$$
= \frac{2}{d} \left\| S^{\otimes 2} - T^{\otimes 2} \right\|_1
\tag{A.51}
$$

$$
\leq \frac{2}{d} \left( \left\| S^{\otimes 2} - S \otimes T \right\|_1 + \left\| S \otimes T - T^{\otimes 2} \right\|_1 \right)
\tag{A.52}
$$

$$
\leq \frac{2}{\sqrt{d}} \left( \left\| S \otimes (S - T) \right\|_2 + \left\| (S - T) \otimes T \right\|_2 \right)
\tag{A.53}
$$

$$
= \frac{2}{\sqrt{d}} \left( \left\| \mathbb{1} \otimes (S - T) \right\|_2 + \left\| (S - T) \otimes \mathbb{1} \right\|_2 \right)
\tag{A.54}
$$

$$
= 4 \| S - T \|_2 \,,
\tag{A.55}
$$

where in we Eqs. (A.51) and (A.54) we used the unitary-invariance of the 1 and 2 norms respectively. In Eq. (A.53) we used that for an operator $T$ on a $d$ dimensional space we have $\|T\|_1 \leq \sqrt{d} \|T\|_2$. So $f$ is 4-Lipschitz continuous; by using Eq. (A.44) and choosing $A, B, C$ and $D$ appropriately we get the result. $\qquad \square$

**Proposition 7.** *The average behaviour of an initial Pauli string, when evolved via a random unitary $U \sim \mu_G$, is to spread uniformly over its connected component, i.e. for Paulis $V$ and $W$*

$$
\mathbb{E}_{U \sim \mu_G} \left| \langle W, UVU^\dagger \rangle_{\mathrm{HS}} \right|^2 = 
\begin{cases}
\frac{d^2}{|C(V)|}, & \text{if } C(V) = C(W), \\
0, & \text{otherwise,}
\end{cases}
\tag{36}
$$

*where $\langle A, B \rangle_{\mathrm{HS}} = \operatorname{tr}\left[A^\dagger B\right]$ denotes the Hilbert-Schmidt inner product.*

*Proof.* After Heisenberg-evolving by some $U \in G$, we have $V \mapsto U^\dagger V U = \sum_W \alpha_W W$. We can find the magnitude of the coefficient $\alpha_W$ by taking the (Hilbert-Schmidt) inner product with $W$. We begin by noting that, by the self-adjointness of the Paulis,

$$
\left| \langle W, UVU^\dagger \rangle_{\mathrm{HS}} \right|^2 = \operatorname{tr}\left[ WUVU^\dagger \right] \operatorname{tr}\left[ WUVU^\dagger \right]^*
\tag{A.56}
$$

$$
= \operatorname{tr}\left[ WUVU^\dagger \right] \operatorname{tr}\left[ W^* U^* V^* U^T \right]
\tag{A.57}
$$

$$
= \operatorname{tr}\left[ WUVU^\dagger \right] \operatorname{tr}\left[ (W^* U^* V^* U^T)^T \right]
\tag{A.58}
$$

$$
= \operatorname{tr}\left[ WUVU^\dagger \right] \operatorname{tr}\left[ UVU^\dagger W \right]
\tag{A.59}
$$

$$
= \operatorname{tr}\left[ WUVU^\dagger \right] \operatorname{tr}\left[ WUVU^\dagger \right]
\tag{A.60}
$$

$$
= \operatorname{tr}\left[ W^{\otimes 2} U^{\otimes 2} V^{\otimes 2} U^{\dagger \otimes 2} \right].
\tag{A.61}
$$

Averaging over $G$ we have

$$\mathbb{E}_{U \sim \mu_G} \left| \langle W, U V U^\dagger \rangle_{\text{HS}} \right|^2 = \mathbb{E}_{U \sim \mu_G} \text{tr}\left[ W^{\otimes 2} U^{\otimes 2} V^{\otimes 2} U^{\dagger \otimes 2} \right] \tag{A.62}$$

$$= \text{tr}\left[ W^{\otimes 2} \Big( \sum_{j,\kappa} Q_{j,\kappa} \frac{\langle Q_{j,\kappa}, V \otimes V \rangle_{\text{HS}}}{\langle Q_{j,\kappa}, Q_{j,\kappa} \rangle_{\text{HS}}} \Big) \right] \tag{A.63}$$

$$= \text{tr}\left[ W^{\otimes 2} \sum_{j,\kappa} \frac{Q_{j,\kappa}}{|C_\kappa| d^2} \Big\langle \sum_{S \in C_\kappa} S \otimes L_j S, V \otimes V \Big\rangle_{\text{HS}} \right] \tag{A.64}$$

$$= \frac{1}{d^2} \text{tr}\left[ W^{\otimes 2} \sum_{j,\kappa} \frac{Q_{j,\kappa}}{|C_\kappa|} \sum_{S \in C_\kappa} \text{tr}[SV] \text{tr}\left[ L_j S V \right] \right] . \tag{A.65}$$

Now, $\text{tr}[SV] = d \delta_{S,V}$, which combined with the term $\text{tr}\left[ L_j S V \right]$ forces $L_j = \mathbb{1}$; we obtain

$$\frac{1}{d^2} \text{tr}\left[ W^{\otimes 2} \sum_{j,\kappa} \frac{Q_{j,\kappa}}{|C_\kappa|} \sum_{S \in C_\kappa} \text{tr}[SV] \text{tr}\left[ L_j S V \right] \right] = \frac{1}{d^2} \text{tr}\left[ W^{\otimes 2} Q_{\mathbb{1}, C_V} \frac{d^2}{|C_V|} \right] \tag{A.66}$$

$$= \frac{1}{|C_V|} \text{tr}\left[ W^{\otimes 2} Q_{\mathbb{1}, C_V} \right] \tag{A.67}$$

$$= \frac{1}{|C_V|} \text{tr}\left[ \sum_{T \in C_V} W^{\otimes 2} T^{\otimes 2} \right] \tag{A.68}$$

$$= \begin{cases} d^2 / |C_V|, & \text{if } C_V = C_W, \\ 0, & \text{otherwise}, \end{cases} \tag{A.69}$$

as desired.

$\square$

**Lemma 8.** *Graph complexity is a $q$-complexity on the component to which the initial string belongs.*

*Proof.* Let us begin by recalling that a $q$-complexity on a time-evolved Pauli string $p_t$ is an expectation value $Q(p_t) = \langle\!\langle p_t | \mathcal{Q} | p_t \rangle\!\rangle$ where $\mathcal{Q}$ is a superoperator satisfying

1. $\mathcal{Q}$ is positive semidefinite, with a spectral decomposition

$$\mathcal{Q} = \sum_a q_a |q_a\rangle\!\rangle \langle\!\langle q_a |, \tag{A.70}$$

   where $q_a \geq 0 \ \forall a$.

2. There exists $M > 0$ such that

$$\langle\!\langle q_a | \mathsf{L} | q_b \rangle\!\rangle = 0, \quad \text{if } |q_a - q_b| > M, \tag{A.71}$$

$$\langle\!\langle q_a | O \rangle\!\rangle = 0, \quad \text{if } |q_a| > M, \tag{A.72}$$

with $\mathsf{L} = [H, -]$ the Liouvillian superoperator. So, we need to find a $\mathcal{Q}$ satisfying these conditions which induces graph-complexity, i.e. $G(p_t) = \langle\!\langle p_t | \mathcal{Q} | p_t \rangle\!\rangle$; we will show that a $\mathcal{Q}$ that does the job is

$$\mathcal{Q} = \sum_{s \in C(p)} \ell(p, s) |s\rangle\!\rangle \langle\!\langle s |, \tag{A.73}$$

where the sum is over Pauli strings $s$ in the same graph component as the initial string $p$. First of all, we have

$$G(p_t) = \sum_s \ell(p,s) |\text{tr}[p_t s]|^2 \tag{A.74}$$

$$= \sum_s \ell(p,s) \langle\!\langle p_t|s\rangle\!\rangle \langle\!\langle s|p_t\rangle\!\rangle \tag{A.75}$$

$$= \langle\!\langle p_t| \left( \sum_s \ell(p,s)|s\rangle\!\rangle \langle\!\langle s| \right) |p_t\rangle\!\rangle \tag{A.76}$$

$$= \langle\!\langle p_t|\mathcal{Q}|p_t\rangle\!\rangle. \tag{A.77}$$

So that $\mathcal{Q}$ reproduces graph complexity. Next we check the conditions on $\mathcal{Q}$, all of which work with $M = 1$:

1. Certainly $q_s = \ell(p,s) \geq 0$.

2. Here we need to check Eqs. (A.71) and (A.72); respectively we have:

   - $\mathsf{L}|s\rangle\!\rangle = |[H,s]\rangle\!\rangle$ is a linear combination of strings $\{s_i'\}_i$ which are nearest neighbours of $s$. But this means that, for any $i$, $\ell(p,s_i') \in \{\ell(p,s), \ell(p,s) \pm 1\}$, so $\mathsf{L}|s\rangle\!\rangle$ is orthogonal to any $|t\rangle\!\rangle$ with $|q_s - q_t| = |\ell(p,s) - \ell(p,t)| > 1$.
   - By assumption we start with a Pauli string so $\langle\!\langle s|p\rangle\!\rangle = 0$ if $\ell(p,s) > 0$, so $M = 1$ continues to work.

We conclude that graph complexity is indeed a $q$-complexity. □

**Lemma 9.** *Krylov complexity constitutes a tight upper bound to the graph complexity.*

*Proof.* We can write the Heisenberg evolved Pauli string $p_t$ as $p_t = \sum_q c_q(t)q = \sum_n d_n(t)O_n$ where $q$ are Pauli strings (normalised so that $\|q\|_2 = 1$) in the same connected component as $p$, and $O_n$ are the Krylov basis operators. We can directly calculate:

$$G(p_t) = \sum_q \ell(p,q)|c_q(t)|^2 \tag{A.78}$$

$$= \sum_q \ell(p,q) |\text{tr}[p_t q]|^2 \tag{A.79}$$

$$= \sum_q \ell(p,q) \left| \text{tr}\left[ p_t \sum_n \text{tr}[O_n q] O_n \right] \right|^2 \tag{A.80}$$

$$= \sum_{n,n'} \text{tr}[p_t O_n]\text{tr}[p_t O_{n'}]^* \left( \sum_q \ell(p,q) \text{tr}[O_n q]\text{tr}[O_{n'} q]^* \right) \tag{A.81}$$

$$\leq \sum_{n,n'} \text{tr}[p_t O_n]\text{tr}[p_t O_{n'}]^* \left( \sum_q \min\{n,n'\} \text{tr}[O_n q]\text{tr}[O_{n'} q]^* \right) \tag{A.82}$$

$$= \sum_{n,n'} \text{tr}[p_t O_n]\text{tr}[p_t O_{n'}]^* \min\{n,n'\}\delta_{n,n'} \tag{A.83}$$

$$= \sum_n |d_n|^2 n \tag{A.84}$$

$$= K(p_t), \tag{A.85}$$

where in Eq. (A.80) we have expanded $q$ in the Krylov basis (note that any component $\widetilde{q}$ of $q$ orthogonal to the Krylov space satisfies $\text{tr}[p_t \widetilde{q}] = 0$). In Eq. (A.82) we have used that a

Pauli string $q$ cannot have any overlap with the $n^{\text{th}}$ Krylov operator unless $n$ is big enough for the commutators to "walk" from $p$ to $q$, which takes at least $\ell(p,q)$ commutations. Finally, in Eq. (A.83) we have used

$$
\begin{aligned}
\sum_q \text{tr}[O_n q] \, \text{tr}[O_{n'} q]^* &= \sum_q \langle\!\langle O_{n'} | q \rangle\!\rangle \langle\!\langle q | O_n \rangle\!\rangle \\
&= \langle\!\langle O_{n'} | \Big( \sum_q |q\rangle\!\rangle \langle\!\langle q| \Big) | O_n \rangle\!\rangle \\
&= \langle\!\langle O_{n'} | \mathbb{1}_{C_p} | O_n \rangle\!\rangle \\
&= \langle\!\langle O_{n'} | O_n \rangle\!\rangle = \delta_{n,n'} \,,
\end{aligned}
\tag{A.86}
$$

where $\mathbb{1}_{C_p}$ is the identity restricted to the component to which $p$ (and $O_n$ and $O_{n'}$) belong. $\square$

**Lemma 10.** *When $t \ll \|H\|_1^{-1}$ (taking $\hbar = 1$) we have $\mathsf{G}(p_t) = \Theta(t^2)$ (excepting the trivial case of $p$ being a symmetry of the dynamics, in which case $\mathsf{G}(p_t) = 0$ for all $t$).*

*Proof.* Employing the notation $\text{Ad}(U) = U(-)U^\dagger$, $\text{ad}(X) = [X,-]$, we have the (Baker-Campbell-Hausdorff) relation $\text{Ad} \circ \exp = \exp \circ \, \text{ad}$, and at short times therefore $p_t = e^{itH} p e^{-itH} = p + it[H,p] + \Theta(t^2)$. From the definition of graph complexity, the fact that $\ell(p,p) = 0$, and using $\mathcal{N}^{(1)}(p)$ to denote the nearest neighbours of $p$, we then have

$$
\mathsf{G}(p_t) = \sum_{q \in C_p} \ell(p,q) \, |\text{tr}[p_t q]|^2
\tag{A.87}
$$

$$
= \sum_{\substack{q \in C_p \\ q \neq p}} \ell(p,q) \, |\text{tr}[p_t q]|^2
\tag{A.88}
$$

$$
= \sum_{\substack{q \in C_p \\ q \neq p}} \ell(p,q) \, \left| \text{tr}\left[ \sum_{j=0}^{\infty} \frac{(it)^j}{j!} \text{ad}_H^j(p) q \right] \right|^2
\tag{A.89}
$$

$$
\approx \sum_{\substack{q \in C_p \\ q \neq p}} \ell(p,q) \, |\text{tr}[pq + it\,\text{ad}_H(p)q]|^2
\tag{A.90}
$$

$$
= t^2 \sum_{q \in \mathcal{N}^{(1)}(p)} |\text{tr}[\text{ad}_H(p)q]|^2
\tag{A.91}
$$

$$
\leq t^2 \left| \mathcal{N}^{(1)}(p) \right| \cdot \|\text{ad}_H(p)\|_1^2 \,.
\tag{A.92}
$$

In truncating the Taylor series we have used

$$
t \, |\text{tr}[\text{ad}_H(p)q]| \leq t \|\text{ad}_H(p)\|_1 \cdot \|q\|_\infty = t \|Hp - pH\|_1 \leq 2t \|Hp\|_1 \leq 2t \|H\|_1 \cdot \|p\|_\infty = 2t \|H\|_1 \ll 1 \,,
$$

where the above line uses Hölder's inequality (twice), the triangle inequality, that the spectral norm of Pauli strings is unity and the assumption $t \ll \|H\|_1^{-1}$. $\square$

Finally, we mention the following simple result:

**Lemma 11.** *Every linear Pauli string symmetry $L$ induces a bijection between an arbitrary component $\{P_i\}$ of the commutator graph and its image $\{LP_i\}$ (where we drop factors of $\pm i$) which are in fact isomorphic as graphs. The converse is false, i.e. there may exist pairwise isomorphic components that are not related via multiplication by any linear symmetry.*

*Proof.* Suppose there is an edge between the vertices $P_i$ and $P_j$, i.e. there exists $H$ in the relevant generating set such that $P_j \propto [H, P_i]$. We then have:

$$LP_j \propto L[H, P_i] = LHP_i - LP_iH = HLP_i - LP_iH = [H, LP_i], \qquad (A.93)$$

where we have used that $L$ is a symmetry of the dynamics, and so commutes with $H$. We conclude that an edge between $P_i$ and $P_j$ implies the existence of an edge between $LP_i$ and $LP_j$; one similarly verifies the converse implication, and concludes that the corresponding graph components (which may be identical) are isomorphic.

To see that the converse is false, we consider the example $\mathfrak{g} = \{XI, YI, ZI, IX, IY, IZ\}$. The commutator graph contains two "triangular components" (corresponding to the vertices $\{XI, YI, ZI\}$ and $\{IX, IY, IZ\}$) which are clearly isomorphic as graphs; they cannot be related by a linear symmetry, however, as the action of the DLA $\mathfrak{g} = \langle i\mathcal{G}\rangle_{\text{Lie}}$ does not admit any (non-trivial) linear symmetries. $\qquad\square$

# B Representation theoretic approach to the module structure and second order symmetries

In this section we analyze the structure of the space of linear operators $\mathcal{L} = \text{End}(\mathcal{H})$ (with $\mathcal{H} = \mathbb{C}^d$ and $d = 2^n$ for $n$ qubits or spin-$\frac{1}{2}$ degrees of freedom) and the characterization of second (and higher) order symmetries from the perspective of representation theory for a number of selected DLAs $\mathfrak{g} \subset \mathfrak{u}(d)$. Following Ref. [26] and our discussion in the main text, these results can be used to make statements about frame potentials and thus about the deviation of uniform DLA ensembles from $k$-designs.

In all our considerations below we are interested in decomposing $\mathcal{L} = \text{End}(\mathcal{H})$ into irreducible representations under the action of a given DLA $\mathfrak{g} \subset \mathfrak{u}(d)$ which can itself be regarded as a Lie subalgebra of $\mathcal{L}$.[10] We will refer to the space $\mathcal{L}$ as the *adjoint module* as it carries a natural adjoint action of $\mathfrak{u}(d)$ by means of the commutator.[11] In fact, since the center of $\mathfrak{u}(d)$ – generated by multiples of the identity – acts trivially, this may equally be interpreted as an action of $\mathfrak{su}(d)$.

The DLA $\mathfrak{g}$ is a Lie subalgebra of $\mathfrak{u}(d)$ and our goal is to decompose the adjoint module $\mathcal{L}$ – and its tensor powers $\mathcal{L}^{\otimes k}$ – into irreducible representations of the DLA $\mathfrak{g}$. This will allow us to identify the number of higher order symmetries. Indeed, the order-$k$ symmetries precisely correspond to the space of trivial representations (singlets) in $\mathcal{L}^{\otimes k}$ (regarded as a module over the DLA $\mathfrak{g}$). As $\mathcal{L}$ corresponds to the complexification of $\mathfrak{u}(d)$ this space can be determined if we understand the branching rules for the DLA embedding $\mathfrak{g} \subset \mathfrak{u}(d)$, specifically the branching of the adjoint representation of $\mathfrak{u}(d)$ on (a complexified version of) itself.

We begin the discussion with the case of the universal DLA $\mathfrak{g} = \mathfrak{su}(d) \subset \mathfrak{u}(d)$ for which we recover well-known results [26]. The detailed description of this familiar case will allow to illucidate the general procedure and philosophy. We will then discuss a selection of other DLAs, while leaving a more comprehensive analysis for a separate publication.

## B.1 Universal case

In the universal case we consider the subalgebra $\mathfrak{su}(d) \subset \mathfrak{u}(d)$ with the adjoint action on $\mathfrak{u}(d)$.[12] As we can think about the latter $\mathfrak{u}(d)$ as the vector space of all (complex) matrices

---

[10] $\mathcal{L}$ can be regarded as a complexification of the real Lie algebra $\mathfrak{u}(d)$.

[11] This should not be confused with the usual adjoint representation which refers to a representation of a Lie algebra $\mathfrak{g}$ on *itself* by means of the commutator.

[12] Here and in what follows the complexification of $\mathfrak{u}(d)$ will be implicitly assumed.

of size $d \times d$, the adjoint module can be regarded as the tensor product $\mathcal{V} \otimes \mathcal{V}^*$, where $\mathcal{V}$ is the fundamental representation of $\mathfrak{su}(d)$ and $\mathcal{V}^*$ is its dual. The decomposition of that tensor product using standard techniques (see, e.g., Ref. [29]) leads to

$$\mathrm{Ad}_{\mathfrak{u}(d)}\Big|_{\mathfrak{su}(d)} = \mathcal{V} \otimes \mathcal{V}^* = 0 \oplus \mathrm{Ad}_{\mathfrak{su}(d)}, \tag{B.1}$$

where Ad refers to the respective Lie algebras (adjoint representation over itself) and 0 denotes the trivial representation. This is just the representation theoretic reflection of the decomposition $\mathfrak{u}(d) = \mathfrak{su}(d) \oplus \mathfrak{u}(1)$, where $\mathfrak{u}(1)$ is generated by multiples of the identity matrix. We stress that the identity matrix has a trivial commutator with all other matrices and thus spans the trivial representation.

Now that we settled some notation it is possible to investigate order-$k$ symmetries. As explained above this amounts to decomposing the $k$-fold tensor product

$$\mathrm{Ad}_{\mathfrak{u}(d)}^{\otimes k}\Big|_{\mathfrak{su}(d)} = \underbrace{\mathfrak{u}(d) \otimes \cdots \otimes \mathfrak{u}(d)}_{k \text{ factors}}\Big|_{\mathfrak{su}(d)} = \underbrace{\mathcal{V} \otimes \cdots \otimes \mathcal{V}}_{k \text{ factors}} \otimes \underbrace{\mathcal{V}^* \otimes \cdots \otimes \mathcal{V}^*}_{k \text{ factors}}. \tag{B.2}$$

On that total space we have two commuting actions of $\mathfrak{su}(d)$ and the permutation group $S_k$. The higher order symmetries correspond to the trivial representations under the action of $\mathfrak{su}(d)$. By Schur-Weyl duality there are two multiplicity-free decompositions

$$\mathcal{V} \otimes \cdots \otimes \mathcal{V} = \bigoplus_\lambda \mathcal{V}_\lambda \otimes \mathcal{S}_\lambda, \qquad \text{and} \qquad \mathcal{V}^* \otimes \cdots \otimes \mathcal{V}^* = \bigoplus_\lambda \mathcal{V}_\lambda^* \otimes \mathcal{S}_\lambda^*, \tag{B.3}$$

into irreducible modules of $\mathfrak{su}(d)$ and $S_k$ (called $\mathcal{V}_\lambda$ and $\mathcal{S}_\lambda$, respectively), where $\lambda$ refers to Young tableaux with $k$ boxes and at most $d$ rows. We note that a singlet with respect to $\mathfrak{su}(d)$ arises precisely (and does so once) by pairing $\mathcal{V}_\lambda$ with its dual $\mathcal{V}_\lambda^*$. This means that the multiplicity space of singlets is given by

$$\bigoplus_\lambda \mathcal{S}_\lambda \otimes \mathcal{S}_\lambda^*. \tag{B.4}$$

If the sum extends over *all* Young tableaux with $k$ boxes, this may be interpreted as the regular representation of $S_k$ and it has dimension $k!$, equal to the size of the group.[13] As the Young tableaux appearing in Eq. (B.4) are limited to precisely $k$ boxes but maximally $d = 2^n$ rows, this assumption leads to the condition $k \leq 2^n$. If this condition is violated one can still compute the number of singlets via Eq. (B.4) as one precisely knows which representations are missing from the sum. However, in concrete applications $k$ is usually small and $2^n$ is very large, so that this is not of practical relevance.

Our derivation gives an alternative perspective on the statements in [26], a perspective that we will soon generalize to situations where a comparably simple notion of Schur-Weyl duality is absent. Yet, the number of higher order symmetries can still be computed through the determination of the multiplicity-space of singlets in tensor products of the form (B.2) or even simply their dimension.

## B.2 Matchgate case

In the matchgate case, the decomposition of the adjoint module was studied in detail using commutator graphs [6], see also the main text and Appendix C. Here we will complement that analysis from a representation theoretic perspective. We find that the latter gives a finer and more versatile description than the graphs.

---

[13]More precisely it is the regular representation of the associated group algebra $\mathbb{C}[S_k]$, regarded as a bimodule over itself.

Table 2: Number of singlets in $\mathcal{L}^{\otimes k}$ (number of order-$k$ higher symmetries) as a function of $n$ for the matchgate DLA $\mathfrak{so}(2n) \subset \mathfrak{u}(2^n)$. The second-last column was predicted using the Online Encyclopedia of Integer Sequences (OEIS).

| $k\mid n$ | 4 | 5 | 6 | 7 | 8 | 9 | 10 | $n$ | OEIS |
|---|---|---|---|---|---|---|---|---|---|
| 2 | 18 | 22 | 26 | 30 | 34 | 38 | 42 | $4n+2$ | A016825 |
| 3 | 330 | 572 | 910 | 1360 | 1938 | 2660 | 3542 | $\frac{2}{3}(4n^3+48n^2+191n+252)$ | A259110 |
| 4 | 9438 | 26026 | 61880 | 131784 | 257754 | 471086 | 814660 | $\frac{16n^6}{45}+\frac{16n^5}{15}+\frac{8n^4}{9}-\frac{11n^2}{45}-\frac{n}{15}$ | A259317 |
| 5 | 368082 | 1769768 | 6852768 | 22535064 | 65211762 | 170263940 | 408493800 | | |
| 6 | 18076916 | 163324304 | 1106722032 | 6018114036 | 27497626310 | 109090538700 | 385005406500 | | |

The matchgate DLA is $\mathfrak{so}(2n)$ which can be identified with $D_n$ in Dynkin's classification of Lie algebras. We follow the conventions of [29] and denote the fundamental weights by $\omega_i$ and the highest root by $\theta$.[14] Comparing the structure of the components of the commutator graph to the representations of $\mathfrak{so}(2n)$ we are led to propose the decomposition

$$\mathcal{L} = 2\mathcal{V}_0 \oplus \bigoplus_{l=1}^{n-2} 2\mathcal{V}_l \oplus 2\mathcal{V}_{\omega_{n-1}+\omega_n} \oplus \mathcal{V}_{2\omega_{n-1}} \oplus \mathcal{V}_{2\omega_n} , \tag{B.5}$$

of the adjoint module. It has been checked using computer algebra for $n = 5, 6, 7$ that these decompositions work and are the only ones that are consistent with the structure of the commutator graph. We note that the decomposition features one more module than there are components in the graph. This is due to the fact that the largest component of dimension $\binom{2n}{n}$ actually decomposes into two irreducible modules $\mathcal{V}_{2\omega_{n-1}}$ and $\mathcal{V}_{2\omega_n}$.

Given the decomposition (B.5) we immediately see that there are two linear symmetries and

$$n \times 2^2 + 2 \times 1^2 = 4n + 2, \tag{B.6}$$

quadratic symmetries, in agreement with Proposition 1. A computer-assisted decomposition of higher tensor products $\mathcal{L}^{\otimes k}$ for various choices of $n$ leads to Table 2.

### B.3 Ising case with arbitrary magnetic field

Our next considerations concern the case $\mathfrak{b}_4$ (in the notation of [9]), which will also allow to deduce statements about the cases $\mathfrak{a}_{13} \cong \mathfrak{a}_{15} \cong \mathfrak{a}_{20}$.

This is the case where the DLA is given by

$$\mathfrak{g} = \mathfrak{su}(\tfrac{1}{2}2^n) \oplus \mathfrak{su}(\tfrac{1}{2}2^n) \oplus \mathfrak{u}(1) \subset \mathfrak{su}(2^n). \tag{B.7}$$

This embedding is known to be maximal (see Table 1 in Ref. [56]) and because the rank of $\mathfrak{g}$ is of the order to the rank of $\mathfrak{su}(2^n)$ we expect relatively simple branching rules. Schematically, the embedding can be understood in terms of the block structure

$$\begin{pmatrix} \mathfrak{su}(\tfrac{1}{2}2^n) & 0 \\ 0 & \mathfrak{su}(\tfrac{1}{2}2^n) \end{pmatrix} \subset \mathfrak{su}(2^n), \tag{B.8}$$

of $2^n \times 2^n$ matrices, with the $\mathfrak{u}(1)$ subalgebra represented by diagonal matrices proportional to

$$\begin{pmatrix} \mathbf{I} & 0 \\ 0 & -\mathbf{I} \end{pmatrix}. \tag{B.9}$$

---

[14]The highest root labels the adjoint representation.

This interpretation will be useful in determining the branching rules for the adjoint module $\mathcal{L}$ when restricting the action to the DLA $\mathfrak{g}$.

A representation of $\mathfrak{g}$ can be labeled by a triple consisting of two representations of $\mathfrak{su}(\frac{1}{2}2^n)$ and one of the abelian Lie algebra $\mathfrak{u}(1)$. Let us denote the adjoint representation of $\mathfrak{su}(\frac{1}{2}2^n)$ by $\theta$ and the fundamental and anti-fundamental representation by $\omega$ and $\bar{\omega}$, respectively. From the matrix embedding we anticipate the decomposition[15]

$$\mathrm{Ad}_{\mathfrak{su}(2^n)}\Big|_{\mathfrak{g}} = (0,0,0) \oplus (\theta,0,0) \oplus (0,\theta,0) \oplus (\omega,\bar{\omega},1) \oplus (\bar{\omega},\omega,-1). \tag{B.10}$$

The first three representations correspond to the space spanned by the matrix in Eq. (B.9) and the two diagonal blocks appearing in Eq. (B.8) while the remaining two representations are associated with the off-diagonal blocks. If we decompose $\mathfrak{u}(2^n)$ instead of $\mathfrak{su}(2^n)$ we will obtain an additional trivial representation (corresponding to the identity operator $\mathbf{I}$ on the total space $\mathcal{L}$), resulting in

$$\mathcal{L}\Big|_{\mathfrak{g}} = 2(0,0,0) \oplus (\theta,0,0) \oplus (0,\theta,0) \oplus (\omega,\bar{\omega},1) \oplus (\bar{\omega},\omega,-1). \tag{B.11}$$

This corresponds to six components of dimensions

$$(1,1,4^{n-1}-1,4^{n-1}-1,4^{n-1},4^{n-1}). \tag{B.12}$$

Indeed, we have

$$\dim(\mathfrak{su}(2^n)) = 4^n - 1, \qquad \dim(\mathfrak{su}(2^{n-1})) = 4^{n-1} - 1, \qquad \dim(\mathfrak{u}(1)) = 1, \tag{B.13}$$

and these are just the dimension of the adjoint representations. The fundamental representations have dimension $2^{n-1}$. The decomposition (B.11) above thus leads to a total dimension of

$$(4^{n-1}-1) + (4^{n-1}-1) + 1 + 2^{n-1}\cdot 2^{n-1} + 2^{n-1}\cdot 2^{n-1} = 4\cdot 4^{n-1} - 1 = 4^n - 1, \tag{B.14}$$

just as expected.

We note that the module structure disagrees with the predictions from the commutator graph. More precisely, in the latter the two representations of dimension $4^{n-1} - 1$ seem to combine into one single component of size $2\cdot 4^{n-1} - 2$. Similarly, the two representations of dimension $4^{n-1}$ are combined into a single component of size $2\cdot 4^{n-1}$. This is due to the fact that there is no appropriate basis of Pauli strings for these irreducible representations.

Using the full result for the decomposition of a single copy of the space of operators we can now also determine the number of quadratic and other higher order symmetries.

The quadratic symmetries should be described by the trivial representations appearing in the decomposition of $\mathfrak{u}(2^n) \otimes \mathfrak{u}(2^n)$ with respect to the DLA in question. If we have the complete representation content of $\mathfrak{u}(2^n)$ (including multiplicities), then it should easily be possible to determine the quadratic symmetries. Indeed, the trivial representation appears precisely (and with multiplicity one) in the tensor product of any representation with its dual. In the case of $\mathfrak{b}_4$, where we have found the decomposition Eq. (B.11), the previous argument leads to

$$\mathrm{End}(\mathcal{H}^{\otimes 2}) = \mathcal{L}^{\otimes 2} = \left[2^2 + 1 + 1 + 1 + 1\right] \times \text{trivial} \oplus \text{others}. \tag{B.15}$$

This suggests that there should be an 8-dimensional space of quadratic symmetries for $\mathfrak{b}_4$.

The other cases as well as higher order symmetries can, in principle, be analyzed in the same fashion. The decompositions replacing Eq. (B.11) will just become significantly more complicated but can easily be computed algorithmically, once the precise branching rules are known. The result is summarized in Table 3.

---

[15]The choice of charge $\pm 1$ for $\mathfrak{u}(1)$ is somewhat arbitrary and depends on the normalization of the generator.

Table 3: Number of singlets in $\mathcal{L}^{\otimes k}$ (number of order-$k$ higher symmetries) as a function of $n$ for the Ising DLA $\mathfrak{su}(2^{n-1}) \oplus \mathfrak{su}(2^{n-1}) \oplus \mathfrak{u}(1) \subset \mathfrak{u}(2^n)$. We note that the results are independent of $n$ as long as we are in a stable regime where $n$ is sufficiently large. Entries that have not yet stabilized are highlighted with brackets. We also note that the table has been transposed in comparison to Table 2.

| $n\mid k$ | 2 | 3 | 4 | 5 | 6 | 7 | 8 | 9 | 10 | 11 |
|---|---|---|---|---|---|---|---|---|---|---|
| 4 | 8 | 48 | 392 | 4192 | 56512 | 917568 | 17227072 | (363276030) | (8445161364) | (213878904780) |
| 5 | 8 | 48 | 392 | 4192 | 56512 | 917568 | 17227072 | 363276032 | 8445161984 | 213879003648 |
| 6 | 8 | 48 | 392 | 4192 | 56512 | 917568 | 17227072 | 363276032 | 8445161984 | 213879003648 |
| 7 | 8 | 48 | 392 | 4192 | 56512 | 917568 | 17227072 | 363276032 | 8445161984 | |

Table 4: Number of singlets in $\mathcal{B}^{\otimes k}$ (number of order-$k$ higher symmetries) as a function of $n$ for the DLA $\mathfrak{su}(2^{n-1}) \oplus \mathfrak{su}(2^{n-1}) \subset \mathfrak{u}(2^n)$ (Case $\mathfrak{a}_{13}$). We observe that there is a slight mismatch with Case $\mathfrak{b}_4$ (which has an additional $\mathfrak{u}(1)$ factor) for small values of $n$ and sufficiently large values of $k$ but that the two results agree in the stable regime.

| $n\mid k$ | 2 | 3 | 4 | 5 | 6 | 7 | 8 | 9 | 10 | 11 |
|---|---|---|---|---|---|---|---|---|---|---|
| 4 | 8 | 48 | 392 | 4192 | 56512 | 917568 | 17227072 | (363276078) | (8445174914) | (213880757716) |
| 5 | 8 | 48 | 392 | 4192 | 56512 | 917568 | 17227072 | 363276032 | 8445161984 | 213879003648 |
| 6 | 8 | 48 | 392 | 4192 | 56512 | 917568 | 17227072 | 363276032 | 8445161984 | 213879003648 |

## B.4 XY model in a longitudinal field

This case is denoted by $\mathfrak{a}_{13}$ in Ref. [9] is similar to the case $\mathfrak{b}_4$, we just need to drop the additional $\mathfrak{u}(1)$ factor and the associated charge-part of the representation in Eq. (B.11). This gives more opportunities for singlets – as we do not insist on representations having $\mathfrak{u}(1)$-charge zero – and the corresponding numbers are summarized in Table 4.

## C  Properties of matchgate commutator graph components

In this section we give an alternative way of computing the graph complexity of a Pauli string in a matchgate circuit. Let $p$ be a Pauli string in component $C_\kappa$. Each vertex in $C_\kappa$ is the product of $\kappa$ distinct Majorana operators (recall Eq. (44)), and so up to sign we can write

$$p = \pm c_{i_1} \dots c_{i_\kappa},$$

where $c_i$ are the Majorana operators and $1 \le i_1 < i_2 < \dots < i_\kappa \le 2n$. Throughout this section we identify a Pauli $p$ with its $\kappa$ constituent Majoranas, which we equivalently identify with a $\kappa$-element subsequence of $(1, \dots, 2n)$. This is equivalent to simply picking a $\kappa$-element subset of $\{1, \dots, 2n\}$, but we generally refer to the subsequence representation $(i_1, \dots, i_\kappa)$ to emphasize the ordering of the Majoranas.

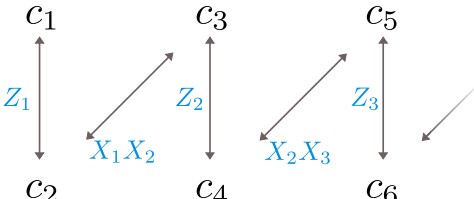

Figure 7: (Taken from [6], Fig. 9(a)) The vertices of this graph represent the Majorana operators $c_j$ for $i \in \{1, \ldots, 2n\}$. The edges (arrows) link pairs of Majoranas that can be mapped from one to the other (up to a scalar) via $c_j \mapsto [c_j, g]$, where $g \in \mathcal{G}$ is a generator, and are labelled with the corresponding generator. As described in the main text, there are two such non-commuting generators for each Majorana (except for $c_1$ and $c_{2n}$ which only have one).

Each Majorana operator anticommutes with exactly two generators in $\mathcal{G} = \{Z_1, \ldots, Z_n, X_1 X_2, \ldots, X_{n-1} X_n\}$ (and thus commutes with everything else, since it is a Pauli string), except for $c_1$ and $c_{2n}$ which anticommute with just one each. Specifically, $c_i$ anticommutes with $Z_{\lceil i/2 \rceil}$ and $X_{\lfloor i/2 \rfloor} X_{\lfloor i/2 \rfloor + 1}$ (with $X_0 X_1 = X_n X_{n+1} = I$ as boundary cases). Furthermore, up to a scalar, the product of the Majorana $c_i$ and one of the two non-commuting generator elements results in either $c_{i+1}$ or $c_{i-1}$, depending on the generator used. These relationships are captured in Fig. 7.

This allows us to describe the neighbours of $p = c_{i_1} \ldots c_{i_\kappa}$ (recalling that the indices are ordered) in the commutator graph in terms of Majorana operators. Let $g \in \mathcal{G}$ be a generator. The operator $g$ anticommutes with either zero, one, or two of the Majoranas that comprise $p$. If it anticommutes with zero, then $g$ and $p$ commute. If it anticommutes with exactly one Majorana $c_i$, then (up to sign)

$$[g, p] = 2i p', \quad \text{where } p' = p c_i c_{i \pm 1},$$

where the $\pm 1$ depends on the precise generator used. If $g$ anticommutes with two Majoranas $c_i$ and $c_j$ with $i < j$, then it follows that $[g, p] = 0$ and $j = i + 1$. Therefore, the neighbours of $p$ in the commutator graph are precisely the Pauli strings $q = c_{j_1} \ldots c_{j_\kappa}$ (up to sign, and recalling that the indices are ordered) such that

$$\sum_{\alpha=1}^{\kappa} |i_\alpha - j_\alpha| = 1. \tag{C.1}$$

This generalises to provide a way to compute distances between arbitrary operators $p, q \in C_\kappa$. A path between $p$ and $q$ corresponds to raising and lowering the index values of the Majorana operators in $p$, one step at a time, to match those in $q$. In particular, since each operator in $C_\kappa$ is the product of $\kappa$ *distinct* Majoranas, all paths between $p$ and $q$ must take the Majorana $c_{i_\alpha}$ to $c_{j_\alpha}$, for all $\alpha \in [\kappa] := \{1, 2, \ldots, \kappa\}$. This implies that

$$\ell(p, q) = \sum_{\alpha=1}^{\kappa} |i_\alpha - j_\alpha|. \tag{C.2}$$

Several small results about the structure of the components $C_\kappa$ follow almost immediately from the above observation. Recall from Eq. (38) that the average over the dynamics of the graph complexity of an initial Pauli string $p$ is given by

$$\mathop{\mathbb{E}}_{U \sim \mu_G} \mathsf{G}(U^\dagger p U) = \frac{1}{|C(\kappa)|} \sum_{q \in C_\kappa} \ell(p, q) = \frac{1}{\binom{2n}{\kappa}} \sum_j \sum_{\alpha=1}^{\kappa} |i_\alpha - j_\alpha|, \tag{C.3}$$

for $p = c_{i_1} \ldots c_{i_\kappa}$, where the sum over $\boldsymbol{j}$ denotes a sum over all $\kappa$-element subsequences $(j_\alpha)_{\alpha=1}^{\kappa}$ of the sequence $(1, \ldots, 2n)$, or equivalently all $\kappa$-element subsets of $[2n] = \{1, \ldots, 2n\}$. One can obtain closed form solutions of Eq. (C.3) for small values of $\kappa$. The case $\kappa = 1$ corresponds to a linear chain of length $2n$, with a vertex for each Majorana and edges between the Majoranas $c_i$ and $c_{i\pm1}$; indeed one can verify that Eq. (C.3) reduces to the result Eq. (39) for a linear chain component. In the $\kappa = 2$ component, vertices $p = \pm c_{i_1} c_{i_2}$ are characterised by a pair of integers $1 \leq i_1 < i_2 \leq 2n$; by Eq. (C.1) we see that there are edges between and only between vertices that are horizontally or vertically adjacent (see Fig. 8). One can also obtain a closed form solution for Eq. (C.3) in this case; indeed:

$$\underset{U \sim \mu_G}{\mathbb{E}} \mathsf{G}(U^\dagger c_{i_1} c_{i_2} U) = \frac{1}{\binom{2n}{2}} \sum_{j} \sum_{\alpha=1}^{2} |i_\alpha - j_\alpha| \tag{C.4}$$

$$= \frac{1}{n(2n-1)} \sum_{j_2=1}^{2n} \sum_{j_1=1}^{j_2-1} \left( |i_1 - j_1| + |i_2 - j_2| \right) \tag{C.5}$$

$$= \frac{1}{n(2n-1)} \left[ \left( \sum_{j_2=1}^{2n} \sum_{j_1=1}^{j_2-1} |i_2 - j_2| \right) + \left( \sum_{j_1=1}^{2n-1} \sum_{j_2=j_1+1}^{2n} |i_1 - j_1| \right) \right] \tag{C.6}$$

$$= \frac{1}{n(2n-1)} \left[ \left( \sum_{j_2=1}^{2n} (j_2 - 1) |i_2 - j_2| \right) + \left( \sum_{j_1=1}^{2n-1} (2n - j_1) |i_1 - j_1| \right) \right] \tag{C.7}$$

$$= \frac{1}{6n(2n-1)} \Big[ i_1(i_1 - 1)(6n - i_1 - 1) + (2n - i_1)(2n - i_1 - 1)(2n - i_1 + 1) \\ + i_2(i_2 - 1)(i_2 - 2) + (2n - i_2)(2n - i_2 + 1)(4n + i_2 - 2) \Big], \tag{C.8}$$

where in Eq. (C.6) we have simply rearranged the order of the sums.

**Lemma 12.** *Let $C_\kappa$, for $\kappa \in \{1, \ldots, 2n\}$, be the components of the commutator graph of a parametrised matchgate circuit. Then*

*(a) the diameter of $C_\kappa$ is $\kappa(2n - \kappa)$,*

*(b) each component $C_\kappa$ has a non-trivial automorphism, given by the map $c_{i_1} \ldots c_{i_\kappa} \mapsto c_{2n-i_\kappa+1} \ldots c_{2n-i_1+1}$.*

*(c) If $p = c_1 \ldots c_\kappa$ or $p = c_{2n-\kappa+1} \ldots c_{2n}$, then $\mathbb{E}_{U \sim \mu_G} \mathsf{G}(U^\dagger p U) = \frac{1}{2} \kappa(2n - \kappa)$.*

*Proof.* Note that for Pauli operators $c_1 \ldots c_\kappa$ and $c_{2n-\kappa+1} \ldots c_{2n}$, it follows from Eq. (C.2) that

$$\ell(c_{2n-\kappa+1} \ldots c_{2n}, c_1 \ldots c_\kappa) = \sum_{\alpha=1}^{\kappa} |2n - \kappa + \alpha - \alpha| = \kappa(2n - \kappa).$$

Now we show that for all other choices $p = c_{i_1} \ldots c_{i_\kappa}$ and $q = c_{j_1} \ldots c_{j_\kappa}$, this forms an upper bound. Since we assume that the Majoranas $c_{i_\alpha}$ and $c_{j_\alpha}$ are ordered lexicographically, it follows that $i_\alpha, j_\alpha \in [\alpha, 2n - \kappa + \alpha]$. This immediately implies that

$$|i_\alpha - j_\alpha| \leq |2n - \kappa + \alpha - \alpha| = 2n - \kappa,$$

for all $\alpha$. This completes the proof of part (a).

Part (b) follows as the given map is an automorphism almost trivially, as all the equal Majoranas will still be equal after taking $2n - i_j + 1$, and the ones that differ by 1 will now differ by 1 in the opposite direction.

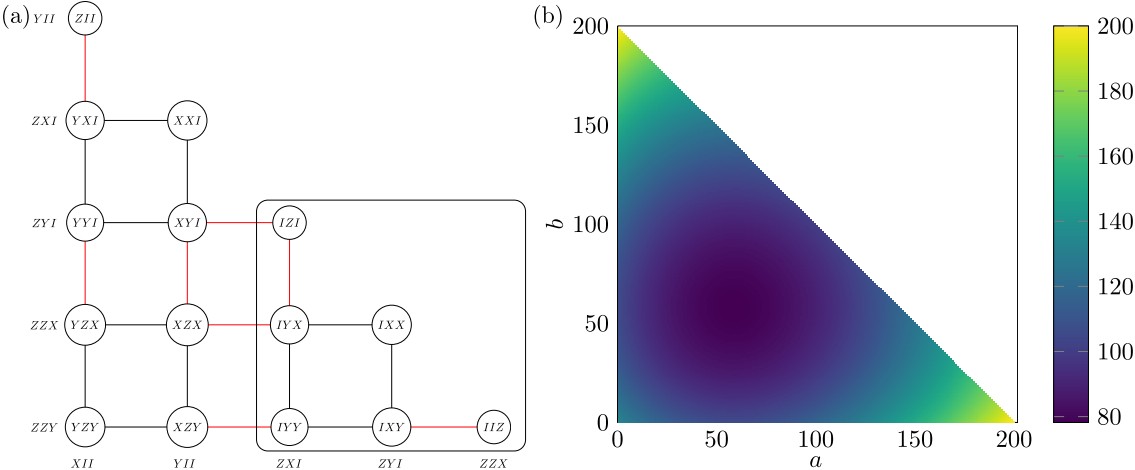

Figure 8: (a) The $n = 3$, $\kappa = 2$ commutator graph component for matchgate circuit dynamics. The Pauli string corresponding to each vertex is (proportional to) the product of two Majoranas. Each row (column) is characterised by the presence of a specific Majorana. Red lines connect nodes related by commutation with $XX$, and black $Z$. The $n = 3$ graph contains the $n = 2$ graph as a subgraph (and more generally the $n = k + 1$ graph contains the $n = k$ graph as a subgraph). (b) The long time average of the graph complexity starting at each node, for $n = 200$.

Now we prove part (c). The average path length to the Pauli string $c_1 \ldots c_\kappa$ (equivalently, to $c_{2n-\kappa+1} \ldots c_{2n}$) is exactly

$$\frac{1}{\binom{2n}{\kappa}} \sum_{\boldsymbol{i}} \sum_{\alpha=1}^{\kappa} |i_\alpha - \alpha| \, .$$

Here the summation is over all sequences $\boldsymbol{i}$ that are strictly increasing $\kappa$-element subsequences of $(1, \ldots, 2n)$. Note that there are precisely $\binom{2n}{\kappa}$ such sequences, and that for each such sequence $i_\alpha \geq \alpha$ for all $\alpha \in \{1, \ldots, \kappa\}$. Thus,

$$\frac{1}{\binom{2n}{\kappa}} \sum_{\boldsymbol{i}} \sum_{\alpha=1}^{\kappa} |i_\alpha - \alpha| = \frac{1}{\binom{2n}{\kappa}} \left[ \sum_{\boldsymbol{i}} \sum_{\alpha=1}^{\kappa} i_\alpha - \sum_{\boldsymbol{i}} \alpha \right] \, .$$

The first summation is the sum over all elements of all sequences $\boldsymbol{i}$. Each element $i \in [2n]$ appears in exactly $\binom{2n-1}{\kappa-1}$ sequences (because choosing such a sequence is equivalent to choosing $\kappa$ unique elements from the set $[2n]$), and thus the first summation is equal to $\binom{2n-1}{\kappa-1} \sum_{i=1}^{2n} i = \binom{2n-1}{\kappa-1}\binom{2n+1}{2}$. Similarly, the second summation is equal to $\binom{2n}{\kappa}\binom{\kappa+1}{2}$. Therefore, the average distance of a vertex in component $C_\kappa$ to the vertex $c_1 \ldots c_\kappa$ is

$$\frac{1}{\binom{2n}{\kappa}} \sum_{\boldsymbol{i}} \sum_{\alpha=1}^{\kappa} |i_\alpha - \alpha| = \frac{\binom{2n-1}{\kappa-1}\binom{2n+1}{2}}{\binom{2n}{\kappa}} - \binom{\kappa+1}{2} = \frac{\kappa(2n+1)}{2} - \binom{\kappa+1}{2} = \frac{1}{2}\kappa(2n - \kappa) \, .$$

This completes the proof. $\qquad\square$

We remark that the diameter of connected components of commutator graphs can be directly related to lower bounds on the number of gates needed to form 2-designs over the corresponding ensemble of unitaries [57]. We can also find expressions for average path lengths between vertices in the commutator graphs of matchgate dynamics, which we give in the following theorem. Let $A(\kappa, n)$ be the average distance between all pairs of Pauli strings in $C_\kappa$.

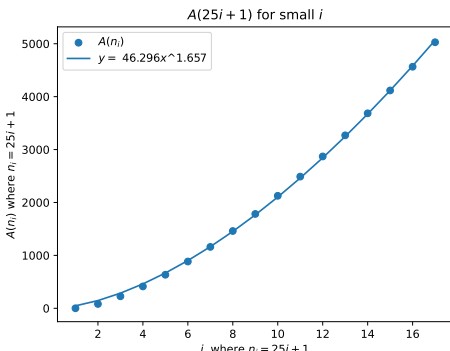
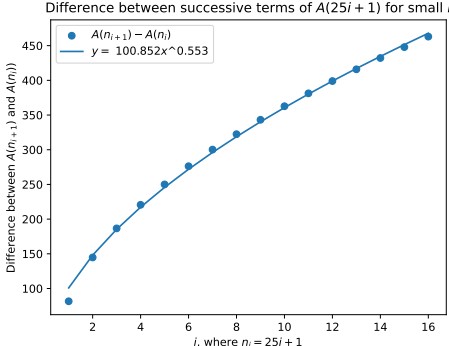

Figure 9: (Left) Plot of $A(n_i) := A(n_i, n_i)$, the average distance between two vertices in component $C_{n_i}$ of the commutator graph of a matchgate circuit on $n_i$ sites, for $n_i = 25i + 1$, $i \in \{1, \ldots, 16\}$. This is plotted alongside a power-law fit provided by `scipy`. The plot suggests that the growth of this function is superlinear but subquadratic in $n$. (Right) Plot of the difference between $A(n_{i+1})$ and $A(n_i)$ $i \in \{1, \ldots, 15\}$. This is plotted alongside a power-law fit provided by `scipy`. If $A(n)$ was linear in $n$, this would be expected to be constant; if $A(n)$ was quadratic this would be expected to be linear.

**Theorem 13.** *Let $C_\kappa$, for $k \in \{1, \ldots, 2n\}$, be the components of the commutator graph of a parametrized matchgate circuit. Then*

$$\binom{2n}{\kappa}^2 A(\kappa, n) = \sum_{p,q \in C_\kappa} \ell(p,q) = \sum_i \sum_j \sum_{\alpha=1}^\kappa |i_\alpha - j_\alpha| \tag{C.9}$$
$$= 2 \sum_{i=2}^{2n} \sum_{j=1}^{i-1} \sum_{\alpha=1}^\kappa \binom{i-1}{\alpha-1}\binom{2n-i}{\kappa-\alpha}\binom{j-1}{\alpha-1}\binom{2n-j}{\kappa-\alpha}(i-j),$$

*where the sums over $\boldsymbol{i}$ and $\boldsymbol{j}$ denote sums over all $\kappa$-element subsequences $(i_\alpha)_{\alpha=1}^\kappa$ and $(j_\alpha)_{\alpha=1}^\kappa$ of the sequence $(1, \ldots, 2n)$.*

*Proof.* It follows from the isomorphism between each Pauli string in $C_\kappa$ and the string of Majoranas $c_{i_1} \ldots c_{i_\kappa}$ that

$$\sum_{p,q} \ell(p,q) = \sum_i \sum_j \sum_{\alpha=1}^\kappa \ell\left(c_{i_1} \ldots c_{i_\kappa}, c_{j_1} \ldots c_{j_\kappa}\right).$$

Applying Eq. (C.2) gives the first equality. To obtain the second equality, consider the number of subsequences $\boldsymbol{i}$ where $i_\alpha = i$ for some value $i \in \{1, \ldots, 2n\}$. There are precisely $\binom{i-1}{\alpha-1}$ choices for the first $\alpha - 1$ entries, and for each such choice there are precisely $\binom{2n-i}{\kappa-\alpha}$ choices for the remaining $\kappa - \alpha$ entries (and exactly one choice for $i_\alpha$, that is, $i$). Applying this gives

$$\sum_{p,q} \ell(p,q) = \sum_{i=1}^{2n} \sum_{j=1}^{2n} \sum_{\alpha=1}^\kappa |i - j|.$$

The claim of the theorem follows by noting that the roles of $i$ and $j$ are symmetric in the above expression and that the summand is zero when $i = j$. □

The summation given in Theorem 13 is rather unwieldy and hard to evaluate in general. Note that it is maximized when $\kappa = n$, and so this bounds from above all instances of the

sum for a fixed $n$. Since $\kappa \leq 2n$ by definition, Lemma 12(a) implies that the diameter of each component is $O(n^2)$, and thus $A(\kappa, n) \leq A(n, n) = O(n^2)$ for all $n$ and $\kappa$. On the other hand, substituting $\kappa = 1$ gives $A(1, n) = \frac{1}{3}n\left(1 - \frac{1}{4n^2}\right)$. It is natural to ask whether the upper bound of $O(n^2)$ is tight or whether the average path length is $O(n)$, or indeed something more nuanced like $O(\kappa n)$.

We give some numerical evidence for small $n$ that none of these are true. Let $A(n) := A(n, n)$ be the value of this summation for $\kappa = n$, which is the value of $\kappa$ that maximizes the sum. We plot the value of $A(n_i)$, where $n_i = 25i + 1$ and $i \in \{1, \ldots, 16\}$ as well as the difference between successive terms in this sequence. These values do not hold particular significance beyond being a small enough computation in aggregate. The plots presented in Fig. 9 indicate that the linear growth of the average path length (with respect to $n$) in all components does not hold as $n \to \infty$, as the differences between successive terms grows with $n$ for the small examples computed. We formalize this in the following conjecture.

**Conjecture 14.** *Let $A(n, n)$ be defined as in Theorem 13, the average distance between two vertices in component $C_\kappa$ of the commutator graph of a matchgate circuit on $n$ sites. Then $A(n, n) = \omega(n)$ and $A(n, n) = o(n^2)$.*

# D   Example graphs

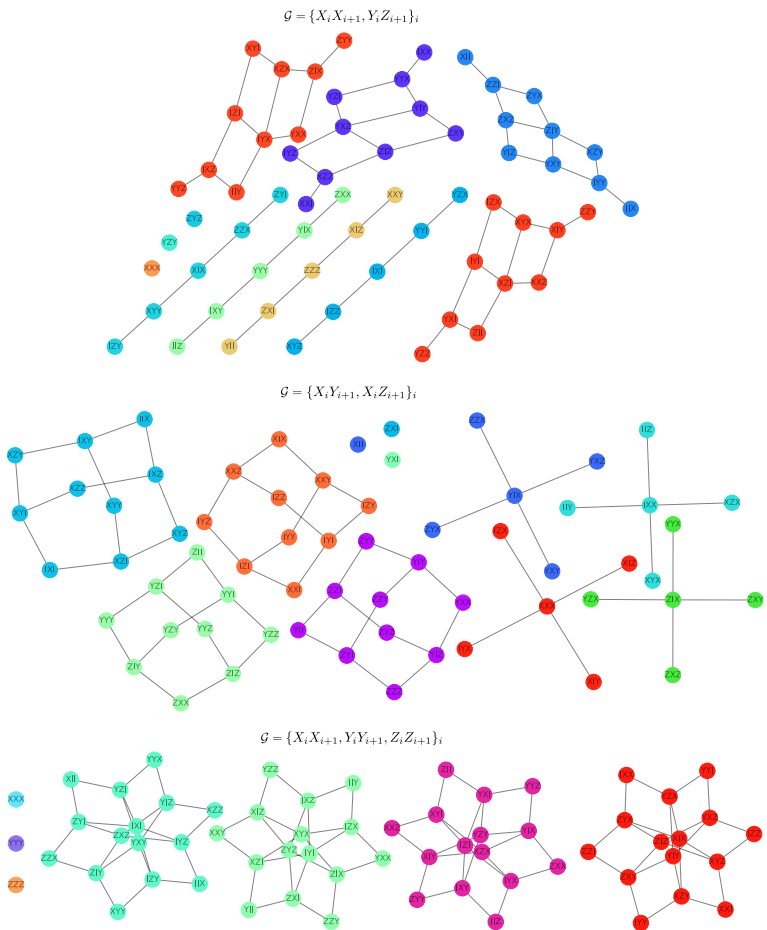

Figure 10: Some further examples of commutator graphs. We omit the identity component, and take as generators the listed elements.

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
