# Peer review of "A graph-theoretic approach to chaos and complexity in quantum systems"

_SciPost Physics Core, doi:SciPost Phys. Core 8, 081 (2025)_

## Round 1 · Referee Report · Roeland Wiersema (Referee 1) · 2025-6-6

Strengths
1- Similar to ref [6], this work tries to beyond the coarse picture of the Dynamical Lie algebra (DLA), and makes statements about the scrambling dynamics of Pauli algebras. The main result is a statement about how averages over Paulis can be calculated I believe that this is a much needed contribution to the field since the DLA view can be quite limiting in making statements about short time dynamics/circuits.
2- The main text requires a good understanding representation theory to understand, but is well written and concise.
Weaknesses
1- I think the results with regards to the Graph complexity could maybe be elaborated on more. For example, I have little intuition for what Lemma 10 is telling me (Similarly for Fig. 6).
Report
I recommend this paper for publication. With regards to the main text, I have some minor questions:
-
With regards to figure 2, U~\mu_G, the Haar measure on G. In the caption however, it says that U is generated as exp(i sum_j c_j H_j) with c_j~Uniform() (I assume ?). This should not give Haar distributed unitaries on G, but some other measure. Could you elaborate on this choice?
-
Could the authors elaborate more on the usage of Lemma 10? What does it mean for the graph complexity to grow quadratically?
-I Am curious if the graph complexity results can be used to make statements about the complexity of Pauli propagation algorithms, whose efficiency relies on truncating the (weighted) commutator graph that is generated by applying layers of a circuit (c.f. [Phys. Rev. A 99, 062337] and arxiv.2306.05400). I feel like there must be direct connections with both the Krylov and Graph complexity. Could the authors comment on this potential connection? If not could the authors comment on if their results could potentially be used to understand the classical simulatibility of quantum dynamics? There is short comment in the conclusion but I am curious if they could elaborate more.
With regards to the proofs, I have the following questions:
1- The proof of Proposition 1 is clear to me, although the last step in A(21) took me a moment to understand. As I understand it, we get a contribution of 1/|C_B1| for each B1, so you get \sum_k \sum_S\inC_k 1/|C_k| = \sum_k where the sum over kappa is again the number of components. Maybe clarify this last step with a sentence below the proof.
2- A(41), the sum over anticommuting elements should probably be in the numerator
3- Proof Proposition 7, The Hilbert Schmidt product is already used in the proof of prop 1 and denoted as <>{HS} instead of <>
Requested changes
1- Page 3, last paragraph, I think "...the same connected component as p (see Fig. 1(c))" should refer to Fig1(a)?
2- Table 1, Ref[9], 2-local Hamiltonians on undirected graphs have also been classified: [arxiv.2409.19797]
3- There is work on "g-sim" from the early 2000s: [arxiv.org.0512209] and [Phys. Rev. Lett. 97, 190501]
4- Fig 3 is not referred to anywhere.
Recommendation
Ask for minor revision

---

## Round 3 · Referee Report · Roeland Wiersema (Referee 1) · 2025-9-25

Report

I want to thank the authors for their detailed response and congratulate them on the outstanding work! I recommend the paper for publication.

Recommendation

Publish (easily meets expectations and criteria for this Journal; among top 50%)

---

## Round 3 · Author Response

Dear Editor,

Thank you for handling our submission. We would also like to thank the Referee both for their time and their helpful suggestions, and for their recommendation that our paper be accepted for publication in SciPost Physics Core

Looking forward to hearing back from you,
The authors

Response to referee comments:

*** Strengths ***

>> 1. Similar to ref [6], this work tries to beyond the coarse picture of the Dynamical Lie algebra (DLA), and makes statements about the scrambling dynamics of Pauli algebras. The main result is a statement about how averages over Paulis can be calculated I believe that this is a much needed contribution to the field since the DLA view can be quite limiting in making statements about short-time dynamics/circuits.

>> 2. The main text requires a good understanding representation theory to understand, but is well written and concise.

We thank the referee for their time, favourable recommendation, and many helpful comments.

*** Weaknesses ***

>> 1. I think the results with regards to the Graph complexity could maybe be elaborated on more. For example, I have little intuition for what Lemma 10 is telling me (Similarly for Fig. 6).

Response: Lemma 10 (and its numerical demonstration in Fig. 6) illustrate the universal behaviour of graph complexity at short timescales, namely, that it always grows quadratically. As we remark in the text, this is in some sense disappointing, as it means that the short-time growth of graph complexity cannot be probing anything interesting. This is for example in contrast to Krylov complexity, the short-time growth of which is a well-studied proxy for quantum chaos in certain systems. We do not believe that there is anything particularly special about the fact that the growth is quadratic (as opposed to any other monomial); indeed inspecting the proof of Lemma 10 one sees that the generalisation to "$k$th-order graph complexity" $\mathsf{G}^{(k)}(p_t) = \sum_q \ell(p,q) |c_q(t)|^k$ would at short times grow like $t^k$. In principle this generalised quantity could be of interest, although it seems to us that $k=2$ (the choice of the text) is by far the most natural due to the 2-norm orthonormalisation of the Paulis.

*** Report ***

>> I recommend this paper for publication. With regards to the main text, I have some minor questions:

>> With regards to figure 2, $U\sim\mu_G$, the Haar measure on $G$. In the caption however, it says that $U$ is generated as $\exp(i \sum_j c_j H_j)$ with $c_j\sim{\rm Uniform}()$ (I assume ?). This should not give Haar distributed unitaries on $G$, but some other measure. Could you elaborate on this choice?

Response: This is a good point which we thank the Referee for raising; we agree that the current wording, which suggests that it is clear \textit{a priori} that the distribution obtained from exponentiating Hamiltonians whose constituent Paulis are weighted with uniformly random coefficients (of magnitude $\mathcal{O}(1)$) should approximate the Haar distribution on the corresponding Lie group, is certainly incorrect. We have reworded this to emphasise that what the figure is showing is numerical evidence that this distribution approximates (at least the second moment of) the Haar distribution, rather than confirming a theoretical prediction that it should. We interpret this as suggesting (if not, admittedly, proving) that our results will with high probability accurate for evolution generated by Hamiltonians that are random in the above sense.

>> Could the authors elaborate more on the usage of Lemma 10? What does it mean for the graph complexity to grow quadratically?

Response: As discussed a little in an earlier response, we view the primary use of Lemma 10 as being to rule out graph complexity as a particularly useful diagnostic of quantum chaos. As also discussed above, the fact that the growth is initially quadratic is simply coming from the "2" in $\mathsf{G}(p_t) = \sum_q \ell(p,q) |c_q(t)|^2$.

>> I am curious if the graph complexity results can be used to make statements about the complexity of Pauli propagation algorithms, whose efficiency relies on truncating the (weighted) commutator graph that is generated by applying layers of a circuit (c.f. [Phys. Rev. A 99, 062337] and arxiv.2306.05400). I feel like there must be direct connections with both the Krylov and Graph complexity. Could the authors comment on this potential connection? If not could the authors comment on if their results could potentially be used to understand the classical simulatibility of quantum dynamics? There is short comment in the conclusion but I am curious if they could elaborate more.

Response: We agree that the graph complexity (and commutator graph properties more generally) seem very closely related to classical simulability, and in particular the efficiency of Pauli propagation algorithms. First of all, the complexity of Heisenberg-evolving some initial Pauli string is evidently upper-bounded by the size of the connected component to which it belongs (as noted in the various ``$\mathfrak{g}$-sim'' papers in the case when the string belongs to the dynamical Lie algebra). Our comment in the conclusion was that one could imagine these Pauli propagation algorithms not so much as truncating the evolving linear combination of Pauli strings "in real time" as the evolution proceeds, but rather as simply taking place on a commutator graph with the high-weight vertices having been removed from the start; an interesting question would then be to what extent one would imagine the removal of certain vertices to fracture components of the graph into manageably-small components that can be readily simulated. This appears to some extent to be captured by the notion of "$k$-connectivity" from graph theory; we would in future be very interested in attempting to calculate such quantities for various graphs of interest. In particular one might hope that the matchgate graph possesses enough structure to make such calculations tractable. We thank the Referee for the thought-provoking questions.

>> With regards to the proofs, I have the following questions:

>> The proof of Proposition 1 is clear to me, although the last step in A(21) took me a moment to understand. As I understand it, we get a contribution of $1/|C_{B_1}|$ for each $ B_1$, so you get $\sum_k \sum_{S\in C_k} 1/|C_k| = \sum_k$ where the sum over kappa is again the number of components. Maybe clarify this last step with a sentence below the proof.

Response: Thanks for the feedback, that is indeed the logic but we have now clarified it a little with an extra sentence as suggested.

>> A(41), the sum over anticommuting elements should probably be in the numerator

Response: Indeed!

>> Proof Proposition 7, The Hilbert Schmidt product is already used in the proof of prop 1 and denoted as $\langle \rangle_{\rm HS}$ instead of $\langle \rangle$

Response: Thanks, we have now made the notation consistent.

*** Requested changes ***

>> Page 3, last paragraph, I think "...the same connected component as $p$ (see Fig. 1(c))" should refer to Fig1(a)?

Response: Thanks for the careful reading! In fact we meant neither a nor c, but Fig. 1(d) at that point; the idea was that we refer to Fig. 1(a) earlier in that sentence when talking about the graph breaking up, and then the Fig. 1(d) bit is saying that a Pauli string not only stays within its component, but actively spreads across it. In any event we certainly shouldn't be referring to Fig. 1(c), and have fixed that.

>> Table 1, Ref[9], 2-local Hamiltonians on undirected graphs have also been classified: [arxiv.2409.19797]

Response: We thank the Referee for the reference and have now added it.

>> There is work on "$\mathfrak{g}$-sim" from the early 2000s: [arxiv.org.0512209] and [Phys. Rev. Lett. 97, 190501]

Response: Thanks, yes we agree we should have added references to these original works rather than simply their later popularisation, which we have now done.

>> Fig 3 is not referred to anywhere.

Response: Indeed, thanks; we have now added a reference.
* * *
We again thank the referee for their time, favourable recommendation, and many helpful comments!

---

## Editorial Decision

published